# Entropy-Monitored Kernelized Token Distillation for Audio-Visual Compression

**Hyoungseob Park**[1][*]  **Lipeng Ke**[2]  **Pritish Mohapatra**[2]  **Huajun Ying**[2]
**Sankar Venkataraman**[2]  **Alex Wong**[1]
[1]Yale University  [2]Amazon AGI

## Abstract

We propose a method for audio-visual knowledge distillation. Existing methods typically distill a student model from the latent embeddings or outputs of a teacher. The former requires matching feature dimensions, if not the same architecture, between teacher and student models while the latter supports any teacher-student pairing, but tends to be less performant. Unlike them, we do not explicitly distill from latent embeddings or outputs, but the pairwise relationships between embeddings across samples for each modality; this is realized as a kernel, which is the crux of our method, "Kernelized Token Distillation (KTD)". Specifically, we tokenize and embed the input for a given modality, and compute the Gram matrix across tokens, from which we distill. As audio and visual modalities afford different information for a task, we adaptively modulate distillation by measuring the entropy of each modality, leading to an Entropy-Monitored Kernelized Token Distillation (EM-KTD) scheme. Our method allows for flexibility in complexity of kernel function to model relationships across tokens, which are selectively distilled to ensure high-fidelity supervision for the student. We evaluate EM-KTD on VGGSound and AVS-Bench, where we use 94% fewer parameters than the teacher while preserving 96.9% in performance for audio-visual event recognition and 96.5% on audio-visual segmentation.

## 1 Introduction

Audio (e.g., microphone) and visual (e.g., camera) sensors are equipped on many everyday devices from laptops and tablets to phones and smart home devices. These two sensor modalities provide complementary information for scene understanding, from disambiguating actions events captured in an image and partitioning the space to localizing particular sound source(s), to direct attention and facilitate interactions. Due to the heterogeneity of the two sensors, models processing visual (RGB image) and audio (mel-spectrogram) data typically employ separate encoders (Gong et al., 2022a; Ma et al., 2024; Gao et al., 2023) for each modality, where performance scales with number of parameters. As audio and visual sensors are often found in edge devices with computational resource constraints, the size of current audio-visual models limits their applicability.

To reduce the computational footprint, we aim to compress these models by distilling knowledge from larger audio-visual models (teachers) into smaller architectures (students) with the goal of preserving model performance. Knowledge distillation (KD) strategies may function in the latent feature space (Zagoruyko & Komodakis, 2016; Heo et al., 2019; Passalis & Tefas, 2018) or the output space (Hinton et al., 2015). The former, operating in latent space, is typically more performant, but requires matching the latent dimensions of teacher and student architectures, while the latter in output space is more widely applicable. To circumvent the inflexibility of latent distillation, one may use a set of projection modules (Fig. 1-a) to map latent features of teachers to the dimensions of students (Kim et al., 2018; Liu et al., 2022b). Yet, this incurs extra parameters and possible uncontrolled effects (Miles et al., 2024) as these projection layers need to be trained to approximate the latent features of the teacher. Furthermore, each sensor captures different information pertaining to a task. In the case where one modality is not informative, e.g., audio-visual recognition of a visually occluded object,

---

[*]This work has been done during Hyoungseob Park's internship in Amazon AGI.

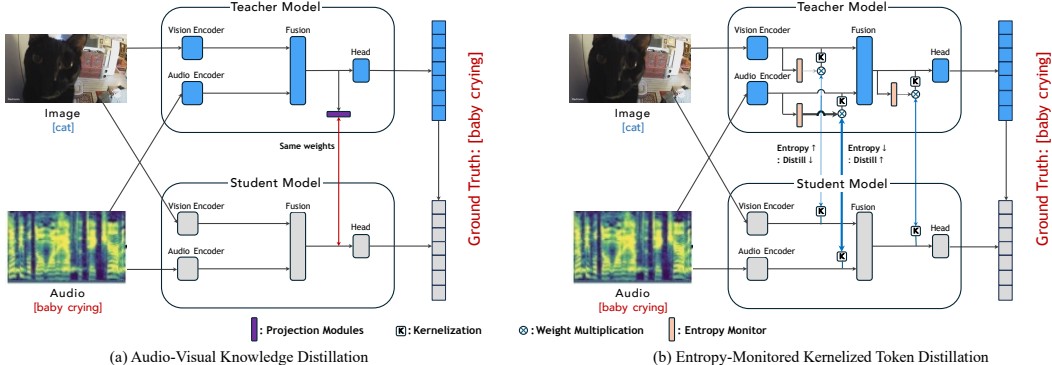

Figure 1: (a) Latent distillation uses projection modules to overcome mismatches in architectures between teacher and student models; latent features are distilled uniformly from teacher to student. (b) Our method selectively distills the latent relationships across data (tokens) and does not assume matching architectures.

distillation can backfire. Hence, our goal is to enable selective distillation, based on the modality's informativeness of the task, that can be applied across student-teacher architectures.

To facilitate audio-visual KD in an architecture agnostic manner, we instead will operate on tokens, customary in Transformers, which are transformed through the feedforward pass. Rather than mimicking the latent embeddings produced by a teacher, which requires matching of dimensions, we distill the relationship between all pairs of latent embeddings. A related work, MTST (Choi et al., 2023), takes a similar approach but is limited to audio only. More importantly, MTST distills from the normalized similarity scores of masked tokens; not only does masking cause a loss of information, but also causes the relationships between tokens of the same example to change depending on which tokens are masked – as their scores are obtained after normalizing over the tokens.

Our work addresses these shortcomings. Inspired by classical kernel methods (Vapnik, 1997), the crux of our method lies in the computation of Gram matrices across latent embeddings. This eliminates the need to match model architecture nor dimensionality and allows for distillation in the latent space. We term this approach "Kernelized Token Distillation" (KTD). It also preserves all of the similarity scores and relationship information across tokens, where each score is meaningful by the metric that one may choose to model such relationships. As a proof of concept, we begin with the simplest function, linear, for kernelization, which amounts to an inner product. We further flexibly increase the complexity of KTD by extending it to polynomial and Radial Basis Function (RBF) kernels, where we compute the relationships between tokens in a higher-dimensional space without the need to explicitly transform them into said space. KTD is versatile and can be applied to distill the (unimodal) latent embeddings of individual modalities as well as teacher multimodal (fused) latent embeddings.

However, as the design of platforms aims to acquire different information from heterogeneous sensors, not all returns for a given modality will be fully informative of the task. Rather than distilling uniformly across all modalities of a teacher, conventional in KD, we will selectively distill based on the informativeness of the latent embeddings of each modality for a given task. This is done by probing the embeddings of each modality and measuring the entropy of its output distribution. Here, entropy serves as a proxy for uncertainty, which acts as an adaptive regularizer to suppress the influence of highly uncertain and non-informative latent embeddings for each modality to ensure a high-fidelity supervision signal. This is realized as an Entropy-Monitored distillation strategy used during the training of the student model (see Fig. 1-b, 2). When used with KTD, this yields an Entropy-Monitored Kernelized Token Distillation (EM-KTD) scheme that monitors each modality to enable selective distillation of informative modalities while being generalizable to different teacher and student architectures. We evaluate EM-KTD on three tasks over two datasets, single- and multi-sound source segmentation on AVS-Bench (Zhou et al., 2022a), and audio-visual event classification on VGGSound (Chen et al., 2020), where EM-KTD preserves 96.9% of teacher model performance with only 6% of the parameters.

**Our contributions**: We propose (1) Kernelized Token Distillation (KTD), a novel knowledge distillation method that allows for flexible metrics to be defined for the kernelization of latent token

embeddings. KTD is also agnostic to architecture, making it suitable for heterogeneous teacher-student pairings. To counter uninformative inputs, we formulate (2) an Entropy-Monitored distillation scheme (EM-KTD) to facilitate learning with KTD, which enables adaptive distillation based on the latent feature entropy of each modality to ensure high-fidelity supervision. (3) We demonstrate EM-KTD for audio-visual tasks across two datasets, where we achieve the state of the art for all tasks.

## 2 RELATED WORK

**Audio-Visual Recognition.** Previous works have studied person identification (Aleksic & Katsaggelos, 2006) and Automatic Speech Recognition (ASR) (Chen & Rao, 1998; Potamianos et al., 2003). Recently, multimodal representation learning audio-visual modality for classification (Ngiam et al., 2011; Ephrat et al., 2018; Kazakos et al., 2019; Xiao et al., 2020; Nagrani et al., 2021; Gong et al., 2022a; Huang et al., 2024a) has drawn significant attention. (Ephrat et al., 2018) separates the speech using audio-visual information (Kazakos et al., 2019) uses a temporal binding window. (Xiao et al., 2020) fuses the audio-visual features in multiple levels of the layers. (Nagrani et al., 2021) proposed a multimodal attention bottleneck. (Gong et al., 2022a) studies contrastive audio-visual masked auto-encoder (Huang et al., 2024a) focuses on self-supervised learning of audio-video representation.

Audio-Visual Segmentation (AVS) aims to segment the pixels with the primary sound source in an image. A line of works (Zhou et al., 2022a; Mao et al., 2023; Gao et al., 2023; Li et al., 2023a; Liu et al., 2023b;a; 2024c; Yang et al., 2024c; Lin & Bertasius, 2024) has tackled the audio visual segmentation in the context of supervised learning. (Zhou et al., 2022a) proposes the AVS-Bench dataset, which is sparsely annotated over the frames. (Mao et al., 2023) uses a variational auto-encoder framework for AVS. (Gao et al., 2023; Li et al., 2023a) uses a cross-attention layer to fuse the audio and vision features. (Liu et al., 2024c) takes advantage of the synthetic data. (Yang et al., 2024c) utilizes a bilateral hint from the foundation model. (Lin & Bertasius, 2024) uses an efficient siamese network to learn audio-visual representation. Further (Liu et al., 2024a) proposes a semi-supervised learning procedure to utilize unlabeled examples via optical flow. In this paper, we specifically utilize transformer-based models, including: CAV-MAE with the different ViT backbone for classification, and UFE-AVS and AVSegFormer (Gao et al., 2023) for audio-visual segmentation, where the models share a similar structure (Gao et al., 2023).

**Knowledge Distillation with Homogeneous Feature Dimensions.** (Hinton et al., 2015) minimizes the KL-divergence output probability distributions between student and teacher. (Tung & Mori, 2019) aims to minimize the feature similarity. (Zagoruyko & Komodakis, 2016; Guo et al., 2023) distills the attention map. (Yim et al., 2017) utilizes the flow of solution procedure (FSP), and (Peng et al., 2019) aligns correlation matrices. (Ahn et al., 2019) maximizes the mutual information. (Heo et al., 2019) distills neuron activation boundary. The previous works utilize multiple feature relations (Park et al., 2019; Tian et al., 2019), where (Park et al., 2019) uses data triplets, while (Tian et al., 2019) uses contrastive learning to transfer the teacher knowledge.

**Knowledge Distillation with Heterogeneous Feature Dimensions.** The most intuitive way to distill information from a teacher model's learned feature distribution is to directly align the features from the teacher and the student (e.g., minimize the distance between their features). This approach becomes infeasible when the teacher and student networks have different feature dimensions. A line of work (Kim et al., 2018; Liu et al., 2022b) addresses this mismatch by introducing the mapping layer to match the feature dimensions, by projecting the feature from the student or teacher model to another model's feature space. Although the loss in these methods minimizes the distance between the student's projected features and the teacher features, the projection layer can be problematic if it learns an overly expressive projection function (Miles et al., 2024), potentially bypassing the distillation by simply mapping the student's features directly to the teacher's features.

Having different feature dimensions can offer greater flexibility in the student architecture. In this context, (Choi et al., 2023) proposes Masked Token Similarity Distillation (MTST), which transfers the normalized distribution of similarity and can be applied to heterogeneous student models. (Choi et al., 2023) argues that applying softmax makes token distillation more stable and scalable by changing it from regression to classification. However, applying softmax discards any linear offset in similarity values, thereby obscuring the teacher information (See Appendix A for further discussion).

Instead, we preserve the teacher's original pairwise relationships across tokens and forgo any masking or normalizing that may corrupt information. This is facilitated by our kernel function, which yields similarity scores (i.e., kernelized tokens) in the same instance from the teacher.

**Knowledge Distillation on Multimodal Models.** Recent advances in multimodal knowledge distillation have introduced effective methods for transferring knowledge from complex, resource-intensive teacher models to compact student models, enabling them to operate efficiently in resource-constrained environments. Existing approaches, such as cross-architecture knowledge distillation, provide flexibility in designing teacher-student pairs; however, these methods are often hindered by the heterogeneity of modality-specific features, particularly when structural differences exist between audio, visual, and textual representations (Deng et al., 2023; Zhou et al., 2024). Previous works (Li et al., 2023b) address the challenge of missing modalities by selectively transferring available information. Nevertheless, these approaches frequently underperform in complex multimodal tasks, as they lack sufficient modality-specific adaptations, leading to diminished accuracy (Wang et al., 2023; Li et al., 2023b). Other methods, such as CRKD (Huang et al., 2024b) and MOMA (Wang & Hatzinakos, 2024), improve modality alignment through adaptation layers but do not dynamically assess the informativeness of each modality, resulting in potential inefficiencies, especially in cases where only certain modalities provide significant context. Liu et al. (2022a) proposed monitored distillation in the output space for depth completion Wong et al. (2020); Wong & Soatto (2021); we propose latent monitored distillation over tokens for generic audio-visual models.

To address these limitations, this paper proposes *Entropy-Monitored Kernelized Token Distillation (EM-KTD)*, a novel distillation framework that integrates *Kernelized Token Distillation (KTD)* for the efficient transfer of cross-modal relationships via similarity matrices and an *Entropy-Monitored* adaptive distillation scheme to reweight modality contributions based on the entropy of each modality. EM-KTD's architecture-agnostic design enables the student model to inherit essential relational structures from the multimodal feature space of the teacher without requiring direct alignment of feature dimensions. Applied to tasks such as audio-visual event classification and segmentation on benchmark datasets like VGGSound and AVS-Bench, EM-KTD demonstrates significant model compression with minimal accuracy loss, offering a promising solution for deployment on the edge in industrial and real-world applications (Gupta et al., 2022; Zheng et al., 2024).

## 3 METHOD FORMULATION

Audio-visual knowledge distillation aims to compress information of a pre-trained teacher model on the target dataset $\mathcal{D}$ to a comparatively smaller (in parameters) student model. We assume a pre-trained teacher model for recognition tasks that takes, as input, multimodal audio-visual data tuple $X = \{M, I\}$ and predicts the probability $\hat{y} \in \mathbb{R}^C$ over a set of $C$ classes $\{c_k\}_{k=0}^{C-1}$. The audio input is converted into a mel-spectrogram $M \in \mathbb{R}^{1 \times B \times F}$, where $B$ refers to the number of mel-frequency bins and $F$ is the number of time frames. We denote $I \in \mathbb{R}^{3 \times H \times W}$ as the associated RGB image. For simplicity, we assume access to both the pretrained teacher model and the target dataset $\mathcal{D} := \{X^{(n)}\}_n^N = \{(M^{(n)}, I^{(n)})\}_n^N$ during the distillation process. The student model is trained on $\mathcal{D}$ under the teacher model guidance, where its latent feature embeddings $f_m^S(X)$ will be guided by those of the teacher, e.g., $f_m^T(X) = \mathbf{z}_m$, for a given modality $m$.

However, as teacher-student dimensions or architecture designs are often mismatched, distilling the output is more amenable for use despite the latent embeddings being more expressive. To address the mismatch in latent feature dimensions between teacher and student models, previous studies have introduced additional projection modules that can cause uncontrolled interactions across dimensions during feature projection (Miles et al., 2024), and a method that distills a similarity matrix of randomly masked and normalized teacher tokens (Choi et al., 2023). However, neither of these approaches is equivalent to directly minimizing the distance between the teacher and student feature representations.

Instead, we propose a method for audio-visual knowledge distillation that is architecture agnostic, but more expressive than methods operating in the output space. Rather than distilling the latent embeddings themselves or modifying the teacher embeddings, we focus on distilling the entire set of relationships across them. To achieve this, we perform a kernelization of latent token embeddings, capturing their pairwise relationships, and minimize the distance between the kernelized tokens of teacher and student models (see Fig. 2 for an overview).

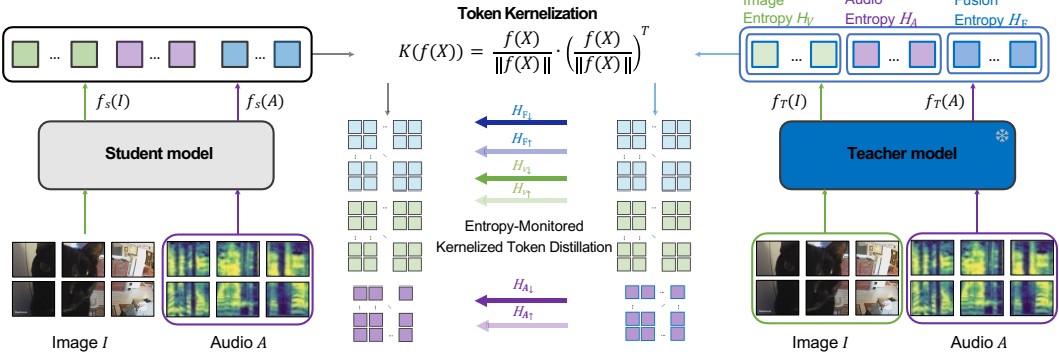

Figure 2: Entropy-Monitored Kernelized Token Distillation. The student (left) and the teacher (right) take in a set of input RGB images and an audio mel-spectrogram. The student distills the kernelized token representations of audio, vision, and fusion modalities from the teacher. The distillation process is guided by the entropy of each modality ($H_v$, $H_a$, $H_F$), obtained from modality-specific classification heads. Lower entropy yields higher distillation weight (see less transparent arrows).

## 3.1 KERNELIZED TOKEN DISTILLATION

Rather than minimizing the distance between the latent embeddings of the student and teacher, we posit that distilling the structure of the teacher's latent space is sufficient to capture its behavior. To this end, we formulate the knowledge distillation problem as an alignment between relationships of data points projected within the teacher's latent space and that of the student. This alignment is achieved by introducing a kernelization step that encodes pairwise relationships for each data point in the teacher's latent space. Akin to kernel methods in classical machine learning, our approach does not require replicating the exact feature vector, but rather the structure of the latent space as discriminative properties come from the separability of points within it.

To this end, we propose a distillation strategy that utilizes a kernel to describe the similarities between points in the latent embedding space. This is done by tokenizing the inputs and encoding them into the latent space of the teacher and that of the student. For the tokens belonging to the same modality of a given example or instance, we compute their kernel, which yields a Gram matrix composed of pairwise similarities and approximates the geometric structure of the latent space. This step is repeated separately for the teacher, student, and for each modality. Amongst the many kernel functions, we choose the simplest one, a linear function, as a proof of concept to demonstrate its efficacy. This method, termed Kernelized Token Distillation (KTD), aims to distill the geometric relationships within the feature space of the teacher to that of the student, giving the student a similar degree of discriminative power as the teacher.

Specifically, for a model, whether teacher or student, we consider the matrix of latent embeddings in the form of token vectors $\mathbf{z}_m \in \mathbb{R}^{N_m \times C}$ from the last transformer encoder block for each modality $m \in \{a, v, f\}$, where $N_m$ is the number of tokens for modality $m$, $C$ is the channel dimension of each token, and $\{a, v, f\}$ refers to audio, vision, and fused modalities, respectively; for ease of notation, we consider the latent fusion embeddings as a separate "fused" modality. For simplicity, we take an example of a linear kernel (for other kernels, see Sec. 3.2). To capture the pairwise relationships of latent embeddings, we compute the Gram matrix $\boldsymbol{\varphi}_m \in \mathbb{R}^{N \times N}$ via a linear kernel across tokens within an instance for a given modality, where its element at index $[i, j]$ is the inner product between the $i$ and $j$-th unit vector, which is normalized latent token embedding vectors from modality $m$, e.g., $\mathbf{z}_m^i$ and $\mathbf{z}_m^j$:

$$\boldsymbol{\varphi}_m[i, j] = {\mathbf{z}_m^i}^\top \mathbf{z}_m^j. \tag{1}$$

We compute $\boldsymbol{\varphi}_m$ for each modality for both teacher and student. We denote the similarity score matrix (captured by any kernel functions) of latent token embedding vectors of the teacher as $\boldsymbol{\varphi}_m^T$ and that of the student as $\boldsymbol{\varphi}_m^S$ for $m \in \{a, v, f\}$. To distill the similarity score matrix for a given modality from the teacher to the student, we enforce the consistency between that of the teacher $\boldsymbol{\varphi}_m^T$

and the student $\boldsymbol{\varphi}_m^S$ by minimizing the Huber loss $\mathcal{L}_{\text{Huber}}(\cdot, \cdot)$ between them:

$$\mathcal{L}_{\text{Huber}}(p, q) \begin{cases} 0.5(p-q)^2, & \text{if } \| p - q \| < 1 \\ |p - q| - 0.5, & \text{otherwise} \end{cases}. \tag{2}$$

This loss is computed for each modality $m \in \{a, v, f\}$ leading to the proposed Kernelized Token Distillation (KTD) loss, $\mathcal{L}_{\text{KTD}}$. Note that the supervision comes from enforcing the consistency across the off-diagonal entries as the diagonal entries (e.g., similarity between the latent embedding and itself) will naturally cancel out:

$$\mathcal{L}_{\text{KTD}} = \sum_m^{a,v,f} \frac{1}{N^2} \sum_{i=0}^{N-1} \sum_{j=0}^{N-1} \mathcal{L}_{\text{Huber}}(\boldsymbol{\varphi}_m^T[i,j], \boldsymbol{\varphi}_m^S[i,j]). \tag{3}$$

We calculate the kernel for every token in each input instance of each modality, rather than for all tokens across instances in the batch. Our choice of instance-wise kernelization is intended to reduce the effect of the quadratic computational complexity. As the loss is computed over all samples and epochs during the course of training, one can still reap the benefits of the latent space structure of the teacher. Note that we distill the teacher token relationships without any modification (i.e., masking or Softmax mapping of the cosine similarity between tokens as in (Choi et al., 2023)), such that the student model's latent space can precisely replicate the geometry of tokens in the teacher latent space.

## 3.2 KERNEL FUNCTIONS

Kernelized Token Distillation (KTD) affords us the flexibility to increase expressivity without adding dimensionality to operate in the high-dimensional latent space. This is facilitated by choosing non-linear kernel functions to capture more complex pairwise relationships across token embeddings. In Sec. 3.1, we considered a linear kernel as a proof of concept; here, we introduce comparatively more complex kernel functions than the linear kernel, which can be applied to KTD for increased performance of distillation.

**Polynomial Kernel** measures a pairwise similarity of vectors through polynomial expansions, which naturally increases the dimensionality and complexity of the function with the growth of $O(C^d)$ for $d$-degree expansion. Yet, as non-linear mapping of kernel methods can be expressed as inner products, this removes the need to operate in high-dimensional space while preserving the benefits of its complexity. Polynomial kernel with degree of $d$ is formulated as:

$$\boldsymbol{\varphi}_m[i,j] = (\mathbf{z}_m^i{}^\top \mathbf{z}_m^j + c)^d, \tag{4}$$

where $c$ is an offset factor, and $\mathbf{z}_m$s are unit vectors.

**Radial Basis Function (RBF).** We further extend our method to the RBF kernel, which can handle non-linear relationships across tokens by mapping the data into an infinite-dimensional space. Similar to the polynomial kernel, this also amounts to (three) inner products (see Eq. 5). The RBF kernel function is denoted as:

$$\boldsymbol{\varphi}_m[i,j] = \exp(-\gamma\|\mathbf{z}_m^i - \mathbf{z}_m^j\|^2) = \exp(-\gamma \cdot \mathbf{z}_m^i{}^\top \mathbf{z}_m^i)\exp(-\gamma \cdot \mathbf{z}_m^j{}^\top \mathbf{z}_m^j)\exp(2\gamma \cdot \mathbf{z}_m^i{}^\top \mathbf{z}_m^j) \tag{5}$$

where $\gamma$ is a hyperparameter that controls the steepness of the function (e.g., a larger $\gamma$ creates a sharper peak and a narrower width, and a smaller $\gamma$ yields a wider width in its decay), which allows flexibility to model the non-linear boundaries between projected data.

## 3.3 ENTROPY-MONITORED DISTILLATION

Audio and visual sensors capture different information. While fusion of the two modalities can provide complementary features to address the weakness of one another, when considered separately, their individual features may not be informative for the task at hand. For instance, an object might be visually occluded, yet its associated sound can be captured within the scene; on the other hand, the object may be visible, but its audibility may be affected by environmental noise. The uninformative features would result in noise in the supervision signal and may possibly degrade the performance of the student model. While one may simply distill from the fused latent embeddings, but in doing so limits the potential of learning the individual strengths of each modality. To enable the teacher to

distill a high-fidelity supervision signal from each modality, we propose Entropy Monitor, an entropy-based approach to facilitate selective distillation. To this end, we employ additional task heads (e.g., a linear layer in classification, denoted as $g_m(\cdot)$) to each teacher encoder branch corresponding to each modality. As the teacher model is frozen, the linear probe allows Entropy Monitor to quantify the entropy corresponding to predictions of the classification heads and adaptively adjust the influence of each sample for each modality during distillation. For instance, in the classification method, the task head $g_m(\cdot)$ is trained to minimize the cross entropy loss calculated with the prediction from the teacher's uni-modal feature, and this allows us to monitor the entropy to adjust the degree of distillation for a given example to regularize the training. The entropy of predictions from each modality is formulated by:

$$H_m(\mathbf{z}_m^T) = -\sum_{c=1}^{C} \sigma(g_m(\mathbf{z}_m^T))[c] \log(\sigma(g_m(\mathbf{z}_m^T))[c]), \tag{6}$$

where $\sigma$ denotes the Softmax operation and $\sigma(g_m(\mathbf{z}_m^T))[c]$ represents the Softmax probability of class $c$ of latent embedding $\mathbf{z}_m$ for modality $m$ of the teacher.

Using the entropy scores, our Entropy Monitor is realized as an adaptive weighting scheme, which dynamically (re-)weights each modality. The weight $w_m$ is given by the negative exponential of the entropy: $w_m = e^{-\lambda H_m(\mathbf{z}_m^T)}$ where $\lambda$ is a hyperparameter that controls the steepness of the function. The calculated weight $w_m$ is utilized to adaptively regularize the loss functions for each modality $m$. Consequently, the final Entropy-Monitored Kernelized Token Distillation loss $\mathcal{L}$ reads:

$$\mathcal{L} = \sum_m^{a,v,f} \frac{w_m}{N^2} \sum_{i=0}^{N-1} \sum_{j=0}^{N-1} \mathcal{L}_{\text{Huber}}(\boldsymbol{\varphi}_m^T[i,j], \boldsymbol{\varphi}_m^S[i,j]). \tag{7}$$

Following the the teacher-student analogy, our Entropy Monitor is akin to a superintendent. Our method, Entropy-Monitored Kernelized Token Distillation, or EM-KTD is minimized along with typical data terms such as cross entropy in the case of classification or segmentation tasks.

**Training Entropy Monitors.** The Entropy Monitor for each modality $m$, is a linear layer $g_m(\cdot)$. We first freeze the teacher model, and the Entropy Monitor will be trained (i.e., linear probing in classification and pixel-wise linear probing in segmentation) before the distillation to the student model. We utilized a cosine annealing schedule following the CAVMAE paper to train $g_m(\cdot)$.

## 4 EXPERIMENTS

For classification, We use CAVMAE (Gong et al., 2022b), fine-tuned on VGGSound and AudioSet-2M datasets and UFE-AVS (Liu et al., 2024a), pre-trained on AVS-Bench for audio-visual segmentation.

**Baselines.** For audio-visual event classification, we evaluated our method using a lower bound, *Vanilla supervised training*, in which the student model is trained from scratch, and five baselines: (1) *Self-supervised pre-training* (SSL-FT) (Gong et al., 2022a); (2) *Hinton's Knowledge Distillation* (KD) (Hinton et al., 2015), which utilizes Kullback-Leibler divergence between probability distributions from the teacher and student models; (3) *Attention Transfer*(AT) (Zagoruyko & Komodakis, 2016), which distills a spatial attention map of features; (4) *Similarity-Preserving Knowledge-Distillation* (SPKD) (Tung & Mori, 2019), which distills the pairwise similarity between *samples*; (5) *Masked Token Similarity Transfer* (MTST) (Choi et al., 2023), which distills of the probability distribution of token similarity. Note: MTST is considered the closest approach to KTD. The main difference in MTST and KTD is the way teacher information is transferred. MTST applies the softmax function to token similarities before distillation and randomly masks them, which ambiguates the teacher token similarities, whereas KTD uses smooth-L1 loss to directly distill the kernelized tokens of the teacher model. A distinction of KTD to MTST is further discussed (Appendix. A). For additional details in the dataset, student model configuration, and evaluation metrics (Appendix E).

For audio-visual segmentation (AVS), we compared our method with four baselines. We adapted the three classification baselines (AT, SPKD, MTST) from audio-visual event classification that can also be applied to segmentation. In particular, for KD, given that the AVS task is a binary classification problem, we replaced the cross-entropy loss with the binary cross-entropy loss. As an additional comparison, we applied the L1 consistency loss between the output features, referred to as L1-consistency in Tab. 2.

Table 1: Comparison with state-of-the-art methods on the VGGSound dataset.

| Method | Model backbone (# Params) | | Acc | mAP | mAUC |
|---|---|---|---|---|---|
| CAVMAE | CAVMAE-ViT-Tiny (10M) | | 54.4 | 53.1 | 96.6 |
| CAVMAE | CAVMAE-ViT-Base (164M) | | 63.9 | 65.0 | 97.9 |
| Method | Teacher model (# params) | Student backbone (# params) | Acc | mAP | mAUC |
| CAVMAE-ViT-Tiny | N/A | ViT-Tiny (10M) | 52.5 | 52.1 | 96.1 |
| KD | CAVMAE-ViT-Base (164M) | ViT-Tiny (10M) | 56.1 | 57.3 | 97.1 |
| AT + KD | CAVMAE-ViT-Base (164M) | ViT-Tiny (10M) | 56.6 | 56.9 | 96.8 |
| SPKD + KD | CAVMAE-ViT-Base (164M) | ViT-Tiny (10M) | 55.6 | 56.1 | 96.6 |
| MTST + KD | CAVMAE-ViT-Base (164M) | ViT-Tiny (10M) | 57.6 | 58.5 | 97.0 |
| KTD + KD (Ours) | CAVMAE-ViT-Base (164M) | ViT-Tiny (10M) | _61.4_ | _62.3_ | _97.6_ |
| EM-KTD + KD (Ours) | CAVMAE-ViT-Base (164M) | ViT-Tiny (10M) | **62.0** | **63.4** | **97.9** |

Table 2: Comparison with state-of-the-art methods on the AVS benchmark. All methods are evaluated on three AVS sub-tasks, including single sound source segmentation (S4), multiple sound source segmentation (MS3).

| Method | | Teacher backbone (# params) | | AVS-Bench-S4 | | AVS-Bench-MS3 | |
|---|---|---|---|---|---|---|---|
| | Methodology | Visual | Audio | $\mathcal{M}_{\mathcal{J}}$ | $\mathcal{M}_{\mathcal{F}}$ | $\mathcal{M}_{\mathcal{J}}$ | $\mathcal{M}_{\mathcal{F}}$ |
| AVSBench-PVTv2 | Supervised | PVTv2-b5 (81.4M) | VGGish (72.2M) | 78.74 | 87.9 | 54.00 | 64.5 |
| DiffusionAVS-PVTv2 | Diffusion | PVTv2 (-) | VGGish (72.2M) | 81.38 | 90.2 | 58.18 | 70.9 |
| AVSegFormer-PVTv2 | Supervised | PVTv2-b5 (81.4M) | VGGish (72.2M) | 82.06 | 89.9 | 58.36 | 69.3 |
| UFE-AVS | Semi-Supervised | PVTv2-b5 (81.4M) | VGGish (72.2M) | 83.15 | 90.4 | 61.95 | 70.9 |
| Method | | Student backbone | | AVS-Bench-S4 | | AVS-Bench-MS3 | |
| | Teacher model | Visual (# Params) | Audio(# Params) | $\mathcal{M}_{\mathcal{J}}$ | $\mathcal{M}_{\mathcal{F}}$ | $\mathcal{M}_{\mathcal{J}}$ | $\mathcal{M}_{\mathcal{F}}$ |
| AVSegFormer | - | PVTv2-b0 (3.4M) | VGGish (72.2M) | 77.41 | 86.76 | 60.45 | 70.83 |
| Vanilla KD (Hinton et al., 2015) | UFE-AVS | PVTv2-b0 (3.4M) | VGGish (72.2M) | 76.86 | 86.29 | 60.49 | 71.20 |
| AT (Tung & Mori, 2019) | UFE-AVS | PVTv2-b0 (3.4M) | VGGish (72.2M) | 77.02 | 85.69 | 58.73 | 69.65 |
| SPKD (Tung & Mori, 2019) | UFE-AVS | PVTv2-b0 (3.4M) | VGGish (72.2M) | 77.59 | 86.25 | 62.09 | 72.96 |
| MTST (Choi et al., 2023) | UFE-AVS | PVTv2-b0 (3.4M) | VGGish (72.2M) | 77.19 | 86.03 | 59.60 | 69.89 |
| KTD (Ours) | UFE-AVS | PVTv2-b0 (3.4M) | VGGish (72.2M) | _79.01_ | _87.26_ | _63.42_ | _74.23_ |
| EM-KTD (Ours) | UFE-AVS | PVTv2-b0 (3.4M) | VGGish (72.2M) | **79.81** | **87.86** | **64.43** | **74.73** |
| AVSegFormer | - | PVTv2-b0 (3.4M) | VGGish-S (18.3M) | 76.11 | 85.61 | 55.50 | 66.65 |
| Vanilla KD (Hinton et al., 2015) | UFE-AVS | PVTv2-b0 (3.4M) | VGGish-S (18.3M) | 77.39 | 85.93 | 53.07 | 63.27 |
| AT (Tung & Mori, 2019) | UFE-AVS | PVTv2-b0 (3.4M) | VGGish-S (18.3M) | 77.21 | 85.59 | 52.93 | 63.93 |
| SPKD (Tung & Mori, 2019) | UFE-AVS | PVTv2-b0 (3.4M) | VGGish-S (18.3M) | 77.09 | 85.46 | 53.56 | 64.13 |
| MTST (Choi et al., 2023) | UFE-AVS | PVTv2-b0 (3.4M) | VGGish-S (18.3M) | 77.04 | 85.05 | 53.04 | 63.83 |
| KTD (Ours) | UFE-AVS | PVTv2-b0 (3.4M) | VGGish-S (18.3M) | _78.14_ | _86.12_ | _58.12_ | _68.56_ |
| EM-KTD (Ours) | UFE-AVS | PVTv2-b0 (3.4M) | VGGish-S (18.3M) | **78.23** | **86.33** | **58.63** | **69.13** |

## 4.1 RESULTS

We note that the teacher models are pretrained and are frozen during the knowledge distillation process, and the student models are trained from scratch.

**Audio-Visual Event Classification.** We evaluated the proposed KTD and UM-KTD with RBF kernels ($\gamma = 0.5$) on the VGGSound dataset, alongside various baseline approaches in Tab. 1. Note: The baselines and the proposed methods utilizes the supervised loss. The result in 1 shows that the proposed Kernelized Token Distillation (KTD) alone improves the three evaluation metrics over the four baselines (KD, AT, SPKD, MTST), with acc of 6.61%, mAP of 3.86%, and 0.85% in mAUC. This result shows that the kernelized token distribution is effective than the output distribution, Adding soft output probability distillation (KD), which distills the teacher information in the output, improves KTD over the baselines by an accuracy of 8.39% and an mAP of 8.59%, which shows that the proposed KTD is synergic with the methods transferring the teacher outputs.

The proposed method, Entropy-Monitored Kernelized Token Distillation (EM-KTD) jointly trained with KD, outperforms the KTD and KD, with an accuracy of 62.0%, mAP of 63.4%, and mAUC of 97.9%. The enhancements are attributed to the Entropy-Monitored Modality Regularization, which adaptively weights the contribution of each modality based on uncertainty measured in the form of the modality's entropy, allowing the student model to prioritize more informative features. The overall results on the VGGSound dataset suggest that the proposed approach effectively distills knowledge from the teacher to a smaller student model (only 6% of the model parameters) maintaining the teacher's performance of 97.02% of accuracy, 97.54% of mAP, and 100% of mAUC.

**Audio-Visual Segmentation.** We evaluated the proposed KTD and EM-KTD with RBF kernels ($\gamma = 0.5$) on the AVSBench dataset for both Single-Sound Source Segmentation (S4) and Multi-Sound Source Segmentation (MS3) tasks. Our method was compared to several baseline approaches,

Table 3: Ablation study on the kernels. RBF outperforms the polynomial and linear kernels.

| Method | Kernel | Acc | mAP | mAUC |
|---|---|---|---|---|
| MTST+KD | Linear | 57.6 | 58.5 | 97.0 |
| KTD | Linear | 60.2 | 59.4 | 97.7 |
| KTD | Polynomial-2 | 60.5 | 60.4 | 97.8 |
| KTD | RBF ($\gamma = 2$) | 60.9 | 61.3 | 97.5 |
| KTD | RBF ($\gamma = 0.5$) | **61.4** | **62.3** | **97.6** |

Table 4: Ablation study of the number of input tokens affected by the input resolution on VGGSound.

| Input Res. | Method | Acc | mAP | mAUC |
|---|---|---|---|---|
| 224×224 | KD | 56.1 | 57.3 | 97.1 |
| 224×224 | AT + KD | 56.6 | 56.9 | 96.8 |
| 224×224 | SPKD + KD | 55.6 | 56.1 | 96.6 |
| 224×224 | MTST + KD | 57.6 | 58.5 | 97.7 |
| 224×224 | EM-KTD | 62.0 | 63.9 | 97.9 |
| 112×112 | KD | 54.5 | 56.5 | 97.1 |
| 112×112 | SPKD + KD | 53.0 | 52.9 | 96.4 |
| 112×112 | EM-KTD | 60.0 | 59.9 | 97.5 |

including KD, AT, SPKD, and MTST. We used mean-Intersection-over-Union ($\mathcal{M}_{\mathcal{J}}$) and F-score metrics to measure segmentation accuracy ($\mathcal{M}_{\mathcal{F}}$).

On the S4 subset, EM-KTD achieved a high mIoU and F-score, retaining 97.1% of the teacher model's performance with significantly fewer parameters. This improvement reflects EM-KTD's effective use of kernelized token distillation to capture essential audio-visual information in a compressed model. The loss of only 2.9% in accuracy while reducing model size demonstrates the strength of our distillation strategy in maintaining quality with reduced complexity.

For MS3 subset, involving segmenting multiple sound sources, EM-KTD showed consistent improvement over the baselines. The adaptive weighting mechanism in EM-KTD, guided by uncertainty-based modality regularization, allowed the model to focus on the most informative audio-visual cues. As a result, EM-KTD outperformed SPKD and Mask, with only 4.5% of the teacher model's visual parameters, showcasing both accuracy and parameter efficiency.

Overall, EM-KTD demonstrated strong performance in both tasks, achieving a balance between high segmentation accuracy and low parameter count. By using kernelized token distillation and adaptive weighting, EM-KTD effectively captures the informativeness of each modality, making it well-suited for efficient, real-world applications where computational resources are limited.

## 4.2 ANALYSIS

**Ablation of Kernel Functions.** The notable advantage of KTD is a freedom in the selection of kernel function. To further explore various kernels, Tab. 3 shows KTD results with the order-2 polynomial and linear kernels and RBF kernel with different hyperparameter ($\gamma = 1$). It is notable that KTD with a linear kernel outperforms MTST, which uses linear kernel as well, by 6.25% on Acc, 6.15% on mAP in audio-visual event classification. The KTD with linear kernel demonstrates the significance of preserving the original kernelized token representation of the teacher during distillation. Also, more complex kernel functions yield better results, but at the cost of computational overhead. The order-2 polynomial kernel requires 2× more matrix multiplications, and RBF kernel needs 3× more multiplications for kernelization. The trade-off is that the order-2 polynomial kernel improves by 0.51%, and RBF kernel ($\sigma = 1$) improves by 2.04%. These results verify the potential of the KTD to be improved with advanced kernel functions.

**Ablation of the Number of Tokens.** Considering the potential limitation of the proposed KTD loss, we evaluate the KTD loss with different numbers of tokens. To decrease the number of the visual tokens to 1/4, we downsampled the student model's input image size by half, where the student and the teacher model use the same config of patch embedding model, where the kernel size is $14 \times 14$ with a stride of 14. The teacher tokens $\mathbf{h}_m^T \in \mathbb{R}^{N \times C}$ are rearranged to their corresponding spatial location, $\mathbf{h}_m^{T\,\prime} \in \mathbb{R}^{h \times w \times C}$, where $h$ and $w$ are the height and the width of the spatial dimension and $N = h \cdot w$. To get the same spatial dimension of the rearranged token to the student model ($\mathbf{h}_m^{S\,\prime} \in \mathbb{R}^{h/2 \times w/2 \times C}$), we use $2 \times 2$ average pool with a stride of 2 for the kernelized token gram matrix, and use the same sized kernelized tokens for the distillation.

As shown in Tab. 4, the downsampled kernelized token distillation outperforms the other baselines presented. This result suggests that distilling downsampled kernelized tokens from the teacher remains effective for the student using lower-resolution input images. This scenario is particularly practical, as the teacher and student models may have different resolutions of visual inputs due to variations in the quality of their optical sensors. Further analysis on the sensitivity study of modality distillation and latency is shown in Appendix C.

**Ablation of the Entropy Monitor $g_m(\cdot)$.** We have conducted an experiment to observe the effect of the Entropy Monitor $g_m(\cdot)$'s architecture. The results are shown in Tab. 5. We observe a marginal

Table 5: Ablation study on the architecture of the Entropy Monitor.

| Method | Architecture | Acc | mAP | mAUC |
|---|---|---|---|---|
| EM-KTD + KD | 3layer MLP | 61.7 | 62.7 | 97.6 |
| EM-KTD + KD | 2layer MLP | 62.0 | 63.3 | 97.9 |
| EM-KTD + KD | 1layer linear | 62.0 | 63.4 | 97.9 |

Table 6: Ablation study of the number of input tokens affected by the input resolution on VGGSound.

| Method | Acc | mAP | mAUC |
|---|---|---|---|
| KTD + KD + Sliding | 61.4 | 62.2 | 97.6 |
| KTD + KD (Ours) | 61.4 | 62.3 | 97.6 |
| EM-KTD + KD + Sliding | 61.9 | 63.3 | 97.9 |
| EM-KTD + KD (Ours) | 62.0 | 63.4 | 97.9 |

difference in the method according to the change. In detail, the MLP architecture consists of the linear layers with non-linear activations (ReLU), with the hidden size of input dimension$\times 2$.

**Ablation Study of the Sliding Window Kernelization.** The downside of the kernelization is the complexity of $O(N^2)$. The complexity for computing a kernel, such as self-attention, is $O(N^2)$. We note that masking applied to MTST does not resolve this issue, as its implementation does not remove tokens before computing the cosine similarity between the features. To reduce the computational cost for the kernelization, we have tried a sliding window approach, akin to Swin Transformers (Liu et al., 2021). This approach kernelizes features within a window shape of $(h/2 \times w/2)$ with the stride of $(h/4, w/4)$, and concatenate them in the batch dimension, reducing the computational cost from $N^2$ to $(\frac{9}{16}N^2)$. We note that we are still comparable in performance. The results with the same teacher-student configuration are shown in Tab. 1 and sliding window in Tab. 6.

**Analysis on Entropy Monitor.** To demonstrate the effectiveness of EM alone, detached from KTD, we conducted the experiment on EM and KD in Tab. 10 of Supp. Mat. We applied EM to Hinton's KD, where EM improved Hinton's KD by 6.71% on VGG sound. While it may seem like EM has less effect when applied to KTD, this is because KTD already preserves 96.9% of the teacher model's performance on VGGSound and 97.1% on S4 benchmarks, which indicates the evaluation metrics are saturated. Hence, it is difficult to push the evaluation metrics further, but we note that EM-KTD still improves the student model consistently over all evaluation metrics. We note that the computation cost comes from calculating entropy is negligible compared to the cost of computing the kernelization, and it filters out the noisy supervision during the distillation to improve performance downstream.

**Discussion on the Applicability of EM to Regression Tasks.** While we focus on classification and segmentation tasks in this paper, EM is not limited to them. In the case of regression tasks, we can re-formulate the objective to estimate uncertainty instead. Specifically, we can consider variance-based uncertainty such as Gaussian negative log-likelihood loss:

$$\mathcal{L} = \frac{1}{2} \left( \frac{(y - \mu)^2}{\sigma^2} + \log \sigma^2 \right). \tag{8}$$

where we can employ a regression head and an uncertainty head, which jointly minimize this objective. In conclusion, rather than using the entropy-based monitor to evaluate the informativeness of each modality, we can model the monitor by the log Gaussian uncertainty, learned by the separate module for each modality.

## 5 DISCUSSION AND LIMITATIONS

We introduced Entropy-Monitored Kernelized Token Distillation (KTD), a novel framework for audio-visual knowledge distillation that compresses large audio-visual models while preserving performance. KTD enables token-level knowledge transfer by entropy-monitored weighting to dynamically prioritize informative modalities (EM-KTD).

Our results on VGGSound and AVSBench show that EM-KTD can achieve 96.9% of the teacher model's performance with just 6% of the parameters outperforming previous state-of-the-art baselines. On segmentation tasks, EM-KTD demonstrates its effectiveness in knowledge distillation with the student model outperforming the teacher model on multi-source object segmentation with only 4% of parameters for the visual encoder. Hence, EM-KTD offers a scalable and resource-efficient solution for audio-visual models aimed at deployment on the edge. An advantage of our formulation is the potential for further improvements through use of advanced kernel functions without incurring the cost of dimensionality. The architectural agnostic nature of EM-KTD also allows for improvements that follow future development of novel (more performant) teacher models, while synergic to the other distillation paradigms (KD, SPKD, and even feature distillation with projector, Appendix. C)

While our experiments focus on audio-visual tasks, the underlying formulation of EM-KTD is modality-agnostic and does not impose restrictions on the type or number of modalities. This flexibility suggests that EM-KTD can extend beyond audio-visual and scale to additional modalities.

## ACKNOWLEDGMENTS

This paper is supported by the Global Industrial Technology Cooperation Center (GITCC) through a grant agreement with the Korea Institute for Advancement of Technology (KIAT), project number P0028922.

## ETHICS STATEMENT

This paper focuses on model compression for audio-visual learning. All experiments were conducted using publicly available and established academic datasets. The study does not involve any personally identifiable human information, and the goal of this work is to create more computationally efficient AI models, which do not introduce new methods for data collection or user identification.

## REPRODUCIBILITY STATEMENT

We provide the methodology in Sec. 3 of the main paper and the implementation details in Appendix E. This will be sufficient to reproduce the results. Furthermore, we will release the code and the pretrained weights.

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

## A    DISCUSSION ON MTST

In the main paper, we argue that applying softmax deforms the original distribution, since softmax discards any linear shift of the logits. We present the proof of how the softmax is a translation-invariant function.

**Lemma (Invariance of Softmax to Translations).**

Let $S \in \mathbb{R}^N$ be a vector with nonzero mean $\mu_S$.

Define $S' = S - \mu_s \mathbf{1}$, where $\mathbf{1}$ is all-ones vector in $\mathbb{R}^N$. Then

$$\sigma(S') = \sigma(S), \tag{9}$$

which demonstrates that the softmax mapping discards the shifts by a constant.

*Proof.* By the definition of softmax function,

$$\sigma(S)[i] = \frac{e^{S[i]}}{\sum_{j=1}^{N} e^{S[j]}} \text{ for } i = 1, ..., N, \tag{10}$$

where $S[i]$ represents $i$-th element in a vector $S$.

We observed that $S'[i] = S[i] - \mu_s$, therefore

$$\sigma(S')[i] = \frac{e^{S'[i]}}{\sum_{j=1}^{N} e^{S'[j]}} = \frac{e^{(S[i] - \mu_S)}}{\sum_{j=1}^{N} e^{(S[j] - \mu_s)}}$$
$$= \frac{e^{-\mu_s} e^{S[i]}}{\sum_{j=1}^{N} e^{-\mu_s} e^{S[j]}} = \frac{e^{S[i]}}{\sum_{j=1}^{N} e^{S[j]}} = \sigma(S)[i]. \tag{11}$$

Since this equality holds for every $i$, we conclude that

$$\sigma(S') = \sigma(S). \tag{12}$$

Thus, $S$ with a constant shift $\mu_S$ to its all components of $S'$ has the same softmax output to $S'$. Therefore, the Softmax operation discards the offset due to its invariance to the translation. $\square$

In this sense, Softmax is not one (logit)-to-one (probability distribution) mapping. Discarding the offset of kernelized representation is critical in the sense that each element is a pairwise cosine similarity, where the change in offset will modify the overall similarity between the tokens. Therefore, we argued that KTD can distill the original kernelized token representation from the teacher, different from MTST.

To further prove the masking strategy is essential to the MTST algorithm, we present the MTST algorithm without masking in Tab. 12, where as our method does not necessitate any random masking.

## B    EXPLORATION OF VARIANTS OF KTD

We explored the variants of Kernelized Token Distillation (KTD) loss. Different from our methods of distilling kernelized tokens within one instance in each modality (audio, visual, fusion), we studied two variants of KTD; (1) Batch-wise Kernelized Token Distillation and (2) Cross-modal Kernelized Token Distillation to justify our choice of instance-wise Kernelized Token Distillation.

**Batch-wise Kernelized Token Distillation (B-KTD).** The kernelized tokens over the batch are useful under the assumption that every cosine similarity between every token is useful for training the student model. Batch-wise Kernelized Token Distillation kernelizes all tokens in each batch, resulting in a single kernel matrix $R^{BN \times BN}$, where $B$ denotes a batch size and $N$ denotes the number of tokens per instance. In contrast, KTD generates $B$ kernelized token matrices with a shape of $R^{N \times N}$.

Other than the underlying assumption, this approach inherits two drawbacks; (1) potential redundancy if the assumption is invalid and (2) increased computational complexity. The kernelization complexity for B-KTD is $O(B^2 N^2)$ compared to KTD's kernelization complexity, $O(BN^2)$, representing a computational overhead increase by a factor of $B$.

Table 7: Sensitivity study on each modality feature with the proposed method and Hinton's distillation on the VGGSound dataset.

| Modality | Kernel | Acc | mAP | mAUC |
|---|---|---|---|---|
| Vision KTD + KD | | 54.5 | 55.1 | 96.6 |
| Audio KTD + KD | Linear | 58.4 | 59.1 | 97.2 |
| Fusion KTD + KD | | 59.6 | 60.3 | 97.4 |
| Full KTD + KD | | **61.2** | **62.1** | **97.7** |

Table 8: Studies on the variants of Kernelized Token Distillation on VGGSound dataset.

| Method | Student | Acc | mAP | mAUC |
|---|---|---|---|---|
| B-KTD | CAVMAE-ViT-Tiny | 59.7 | 60.1 | 97.7 |
| C-KTD | CAVMAE-ViT-Tiny | 61.0 | 61.4 | 97.5 |
| KTD (Ours) | CAVMAE-ViT-Tiny | 61.2 | 62.1 | 97.7 |

**Cross-modal Kernelized Token Distillation (C-KTD).** C-KTD is valid if the cross-modal token similarities before fused are crucial for distillation. The cross-modal kernelization could be denoted as:

$$K(f_v(X_v), f_a(X_a)) = \frac{f_v(X_v)}{|f_v(X_v)|} \cdot \left(\frac{f_a(X_a)}{|f_a(X_a)|}\right)^T,$$

where $X_v, X_a$ indicate the visual and audio input, $f_v, f_a$ refers to the visual and audio encoder. We add the cross-modal kernelized token consistency loss to the KTD losses for each modality, which adds $O(N_v N_a)$ of computational complexity to KTD. The results are shown in Tab. 8.

B-KTD exhibits 2.46% and 3.21% of performance degradation in accuracy and mAP respectively compared to KTD, demonstrating the effectiveness of KTD in achieving better performance with $B$ times less computational overhead compared to B-KTD. The performance degradation observed in B-KTD can be attributed to redundancy, which suggests that the cosine similarity between tokens from different instances is less significant than the similarity between tokens within the same instance.

C-KTD shows a performance degradation of 0.4% in accuracy and 1.2% in the mean average precision (mAP) metric compared to KTD. Given that C-KTD uses additional cross-modal kernelized tokens, this result suggests that the cosine similarity among samples within the same modality has more influence on the distillation than the cross-modal token similarity, where cross-modal kernelized token could work as a noise in distillation. Furthermore, these results suggest the multimodal feature by the fusion encoder is crucial, as in the KTD loss, compared to the similarity between tokens of different modalities before the fusion.

## C  FURTHER ANALYSIS

**Sensitivity study on modality distillation.** We studied the KTD loss of uni-modal and multi-modal features to measure the sensitivity of the student model to the target dataset. In Tab. 7, we ablate the audio, visual tokens (unimodal), and fusion tokens (multimodal) on the VGGSound dataset. We used KTD loss on each set of modality tokens (including audio, visual, and fusion modality). The result with Fusion token KTD loses 2.6% of the original KTD, applied for the three modalities, showing minimal performance degradation among the three modalities. This may appear evident, as the fusion tokens are directly involved in the classification. The audio token KTD shows 4.5% of performance degradation compared to full-modality KTD, whereas the visual token KTD loses 10.9% of the original accuracy.

**Analysis on Latency.** Inference speed of audio-visual event classification on VGGSound with input shapes of image: 3×224×224, audio:1024×128) input on an NVIDIA A10G 24GB GPU: The teacher model takes 9.5ms, while the student model takes 1.5ms, an 82% reduction.

Table 9: Teacher and Student model FLOPs and number of parameters.

| Model | Visual Encoder | | Audio Encoder | | Encoder | |
|---|---|---|---|---|---|---|
| | # Params (M) | FLOPs (G) | # Params (M) | FLOPs (G) | # Params (M) | FLOPs (G) |
| UFE-AVS (T) | 81.44 (1.00×) | 11.76 (1.00×) | 72.14 (1.00×) | 0.86 (1.00×) | 153.58 (1.00×) | 12.62 (1.00×) |
| AVSegFormer (S) | 3.41 (0.042×) | 0.57 (0.048×) | 72.14 (1.00×) | 0.86 (1.00×) | 75.55 (0.492×) | 1.43 (0.113×) |
| AVSegFormer (S-small) | 3.41 (0.042×) | 0.57 (0.048×) | 18.26 (0.253×) | 0.81 (0.94×) | 21.67 (0.141×) | 1.38 (0.109×) |
| CAVMAE-ViT-Base (T) | 78.03 (1.00×) | 15.91 (1.00×) | 78.03 (1.00×) | 44.33 (1.00×) | 156.06 (1.00×) | 60.25 (1.00×) |
| CAVMAE-ViT-Tiny (S) | 4.91 (0.063×) | 1.12 (0.07×) | 4.91 (0.063×) | 3.60 (0.081×) | 9.82 (0.063×) | 4.72 (0.078×) |
| CAVMAE-ViT-Ex-Tiny (S) | 2.68 (0.034×) | 0.61 (0.038×) | 2.68 (0.034×) | 1.97 (0.044×) | 5.36 (0.034×) | 2.58 (0.043×) |

Table 10: Ablation study on kernelized token distillation in different layers on VGGSound.

| Method | Acc | mAP | mAUC |
|---|---|---|---|
| KTD (First) | 56.4 | 57.6 | 97.2 |
| KTD (Middle) | 57.9 | 58.4 | 97.4 |
| KTD (Last) | 60.2 | 59.4 | 97.7 |

Table 11: Evaluation results with different kernels on AVSBench.

| Method | Kernel | S4 | | MS3 | |
|---|---|---|---|---|---|
| | | $\mathcal{M_J}$ | $\mathcal{M_F}$ | $\mathcal{M_J}$ | $\mathcal{M_F}$ |
| MTST | Linear | 77.19 | 86.03 | 59.60 | 69.89 |
| KTD | Linear | 78.96 | 86.99 | 63.30 | 74.09 |
| KTD | RBF ($\gamma = 2$) | 79.01 | 87.26 | 63.42 | 74.23 |

**Analysis on computational complexity.** To compare the computational cost of the teacher and the student model, we present the teacher and the student model's computational complexity in GFLOPs per instance in Tab. 9.

In Audio-Visual segmentation, the student visual encoder maintains more than 96% of original teacher performance on the S4 task and outperforms on the MS3 task with only 4.18% of parameters and 4.84% of computation of the visual encoder, as shown in the main paper. In the context of Audio-Visual event classification, the student model retains 97% of the teacher model's performance on the VGGSound dataset while using only 6.3% of the parameters and 7.8% of the FLOPs of the teacher's combined audio and visual encoders. This highlights the model's capacity to maintain high performance with significantly reduced computational complexity.

**Analysis on the entropy of each modality.** To analyze the proposed Entropy-Monitored strategy's capability, we visualize the entropy of each modality, and the accuracy of samples within specified entropy ranges in Fig. 3.

The Fig. 3-(a) shows the entropy distributions for fusion, audio, and vision modalities (left column) and the accuracy trends across different entropy ranges (right column). The histograms reveal that lower entropy values are concentrated, especially for the fusion modality, indicating high-confidence features crucial for effective knowledge transfer.

The accuracy plots Fig. 3-(b) show a decline in performance as entropy increases for all modalities, confirming the importance of managing modality-specific uncertainty. The fusion modality, with its strong link between low entropy and high accuracy, emerges as a key contributor to the distillation process. By adjusting weights based on entropy, EM-KTD prioritizes reliable modality features, improving the student model's performance.

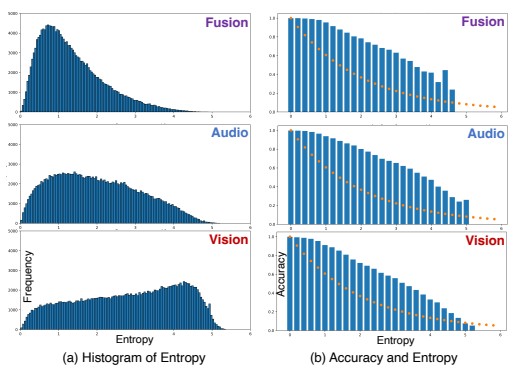

Figure 3: Entropy Analysis. (a) Histogram of the entropy distribution for each modality feature. (b) Accuracy of samples within specified entropy ranges, with orange dots representing the weight of each modality's distillation loss.

**Synergy between KTD and other distillation strategies.** We evaluated SPKD and direct feature distillation (Proj), and incorporated it into KTD to test if it is synergistic with Proj and SPKD. The result indicates that the KTD and

**Inference speed.** Inference speed with input size of (image: $3 \times 224 \times 224$, mel-spectogram: $1024 \times 128$) on an NVIDIA A10G 24GB GPU: The teacher model takes 9.5ms, while the student model takes 1.5ms, an 82% reduction on VGGSound.

**Study on Entropy-Monitored Distillation.** In Tab. 13, we present the proposed Entropy-Monitored distillation can improve not only token-based (EM-KTD), but also output-based distillation objectives. We applied our entropy-monitor to Hinton's Knowledge Distillation (EM-KD), and EM-KD improves KD over 6.7% on Acc, 4.1% on mAP, which demonstrates the effectiveness of Entropy-Monitored Distillation on both distillation objectives, not only pairwise relationship of projected data points, but also distillation objectives applied to output space.

**Different level of distillation layer.** In the paper, we distill the kernelized tokens of the last layer of each modality encoder. To justify this, we conduct the experiments on KTD with kernelized tokens from various level of layers, as shown in Tab. 10. As we distill the later layer representation (First→Middle→Last), the student model improves further. This result shows that the complexity of

Table 12: Additional evaluation results on VGGSound.

| Method | Acc | mAP | mAUC | Method | Acc | mAP | mAUC |
|---|---|---|---|---|---|---|---|
| CAVMAE-ViT-Tiny | 52.5 | 52.1 | 96.1 | KD + Beyer et al. (2022) | 56.3 | 58.3 | 97.3 |
| MTST (None-masked) | 51.3 | 50.3 | 96.0 | MTST + Beyer et al. (2022) | 57.2 | 58.6 | 96.8 |
| KD | | 56.1 | 57.3 | 97.1 | KTD + Beyer et al. (2022) | 61.4 | 61.8 | 97.7 |
| EM-KD | 59.9 | 59.7 | 97.7 | KTD | 60.2 | 59.4 | 97.7 |
| Proj+KD | 53.4 | 53.7 | 96.5 | EM-KTD | 62.0 | 63.4 | 97.9 |
| VKD (Miles et al., 2024) | 61.2 | 61.3 | 97.8 | | | | |

kernelized tokens influences the student model's performance. In particular, tokens distilled from deeper layers—where added non-linearity increases complexity—provide better performance.

**Results various kernels.** As shown in Tab. 11. We compared the KTD with linear kernel to MTST which also uses linear kernel. KTD with linear layer improved 2.29% on $\mathcal{M}_{\mathcal{J}}$ and 1.1% on $\mathcal{M}_{\mathcal{F}}$ in S4 task, and 6.21%, 6.01% respectively in MS3. This results demonstrates the effectiveness of KTD's distillation strategy that preserves the teacher token similarities than normalize and randomly mask.

# D    ADDITIONAL RESULTS

**Ablation of Entropy-Monitor** (Left Column, Row 3-4, Tab. 12). We ablate the entropy-monitor along with Hinton's KD. The entropy is derived from each instance's predicted probability distribution over the classes and adjust the distillation loss weight.

**Combination with the augmentation strategy** (Tab. 12, Right, Rows 1-3). Beyer et al. (2022) proposed one would just need to apply a normal distillation function but simply train for longer instead of opting for more complicated distillation methods. To clearly show the benefit of KTD, we present the experiment of KD, MTST, and KTD with Beyer et al. (2022).

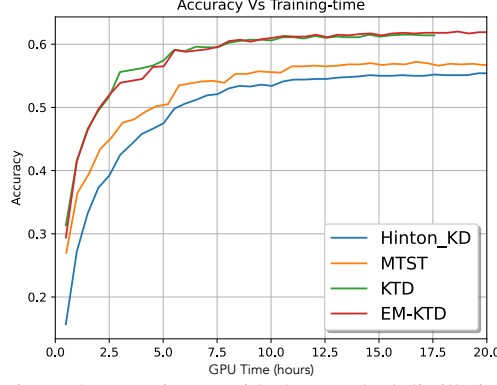

Figure 4: Experiment with the matched distillation time in GPU hours.

**Comparison of the baselines with the matched training GPU time.** Fig. 4 shows the experiment with the matched gpu distillation training time of Hinton's KD, MTST, KTD, and EM-KTD. KTD and EM-KTD outperforms KD and MTST with the same distillation GPU hours, which demonstrates the effectiveness of our distillation strategy.

**Comparison with Projection-Based Feature Distillation** (Tab. 12, Left, Rows 5-7). To investigate how a projection layer's expressiveness affects distillation, we compare EM-KTD with two projection-based methods: a naive baseline (Proj+KD) and the state-of-the-art VKD (Miles et al., 2024). The results demonstrate that EM-KTD surpasses even VKD, highlighting the strength of our projector-free kernelized token distillation approach.

# E    FURTHER EXPERIMENT DETAILS

**Dataset.** We evaluate our distilled student model on two tasks: *audio-visual event classification* and *audio-visual segmentation*, following the dataset splits and evaluation protocols of prior works (Gong et al., 2022a; Liu et al., 2024a). For audio-visual event classification, we use the VGGSound dataset (Chen et al., 2020), which contains 10-second video clips across 200,000 samples. After excluding unavailable YouTube videos, we utilized 182,536 samples for training and 15,331 samples for testing. For audio-visual segmentation, we evaluate on the AVSBench-Object dataset (Zhou et al., 2022c), a benchmark with pixel-level annotations for sound source localization, which includes two sub-tasks Single-Sound Source Segmentation (S4) and Multi-Sound Source Segmentation (MS3). We tokenize image and audio inputs as outlined in (Gong et al., 2022a; Liu et al., 2024a), ensuring consistency with existing benchmarks and fair evaluation of the distillation strategy.

Table 13: Study on the student model with less layer depth $(11 \rightarrow 6)$ on the VGGSound dataset.

| Method | Model backbone (# Params) | | | Acc | mAP | mAUC |
|---|---|---|---|---|---|---|
| CAVMAE | ViT-Ex-Tiny (8M) | | | 54.5 | 54.0 | 96.4 |
| CAVMAE | CAVMAE-ViT-Base (164M) | | | 63.9 | 65.0 | 97.9 |
| Method | Teacher model (# params) | Student backbone (# params) | | Acc | mAP | mAUC |
| CAVMAE-ViT-Ex-Tiny | N/A | ViT-Ex-Tiny (8M) | | 52.1 | 51.8 | 96.0 |
| KD | CAVMAE-ViT-Base (164M) | ViT-Ex-Tiny (8M) | | 55.1 | 57.0 | 97.2 |
| AT + KD | CAVMAE-ViT-Base (164M) | ViT-Ex-Tiny (8M) | | 56.9 | 57.5 | 96.8 |
| SPKD + KD | CAVMAE-ViT-Base (164M) | ViT-Ex-Tiny (8M) | | 54.4 | 54.9 | 96.4 |
| MTST + KD | CAVMAE-ViT-Base (164M) | ViT-Ex-Tiny (8M) | | 57.0 | 57.9 | 96.9 |
| KTD (Ours) | CAVMAE-ViT-Base (164M) | ViT-Ex-Tiny (8M) | | 60.2 | 61.5 | 97.3 |
| KTD + KD (Ours) | CAVMAE-ViT-Base (164M) | ViT-Ex-Tiny (8M) | | 61.0 | 62.4 | 97.6 |
| EM-KTD + KD (Ours) | CAVMAE-ViT-Base (164M) | ViT-Ex-Tiny (8M) | | 61.2 | 63.2 | 97.9 |

**Student model architecture.** We use the ViT-base (Vaswani et al., 2017) (167.8M parameters) based CAV-MAE++ model, finetuned on VGGSound as the teacher model. For the student model we replace the ViT-base encoders with ViT-tiny, which only takes (10.0M parameters, 6% of teacher model). For segmentation, the teacher UFE-AVS model uses a PVTv2-B5 (Wang et al., 2021) based visual encoder (81.44M parameters) which is replaced with PVTv2-B0 (3.7M parameters, 4.5% of teacher model) for the student model. We replaced the decoder to match the feature dimension of PVTv2-B0 architecture channels (46.0M $\rightarrow$ 32.3M parameters).

**Augmentation strategy.** We followed the augmentation strategy of the original teacher models' augmentation for the student model distillation.

**Runtimes.** The model with KTD on VGGSound takes 6.75 hours on a single A100 GPU, where the model with Hinton's KD and MTST takes 3.25 and 6.45 hours on a single A100 GPU.

**Hyperparameters.** To ensure reproducibility, the hyperparameters for our proposed method and baseline models are released as follows. On the *AVSBench* datasets, KTD is trained with learning rate of 2.5e-5 with a loss weight of 12. For the models trained from scratch and knowledge distillation (KD) baselines, a learning rate of 2e-5 was used. The loss balance factors for the KD baselines were set as follows: KD at 300, AT at 1000, SPKD at 2000, and MTST at 3000. These settings helped ensure optimal performance and comparability across methods. For the VGGSound dataset, we use an initial learning rate of 2e-4 when training from scratch and 1e-4 when finetuning the CAV-MAE pretrained model. For KD based training on the VGGSound dataset, we use the following (learing rate, distillation loss weight) setting: (2e-4, 1000) for vanilla-KD, (1e-3, 3333) for AT, (5e-4, 666) for SPKD, (5e-4, 333) for MTST, (5e-4, 333) for KTD, (1e-3, 666) for EM-KTD. The KD loss weights are relative to ground-truth label-based CE loss weight.

**Evaluation metrics.** To evaluate the audio-visual classification results, we use Accuracy (Acc), mean Average Precision (mAP), and mean Area Under the Receiver Operating Characteristic Curve (mAUC) following (Gong et al., 2022a). For audio-visual segmentation, We adopt the mean Intersection-over-Union ($\mathcal{M}_{\mathcal{J}}$) and F-score ($\mathcal{M}_{\mathcal{F}}$) as our evaluation metrics following previous methods (Zhou et al., 2022b; Gao et al., 2023).

**Teacher-Student model configuration.** Details on the architectural difference between the teacher and the student model are shown in Tab. 14. The teacher and student architectures are totally different which can only be covered by MTST (Choi et al., 2023) and SPKD (Tung & Mori, 2019) among the previous knowledge distillation pipelines, without introducing additional projection layers to match the dimensions.

**Clarification on AVSBench training set.** We clarify that the labeled data points are utilized, which is the same data splits used to train the previous supervised Audio-Visual Segmentation (teacher) methods. This is different from that of the UFE-AVS teacher model, which uses unlabeled data points with preprocessed optical flows for auxiliary information.

**Baseline details.** We provide further details for the baseline methods in here.

- *Vanilla supervised training*. The student model is trained from scratch. This setting is to be considered as a lower-bound on the performance of the student model.

Table 14: *Specifications on the teacher and the student model architectures.* # Heads denotes the number of heads, # Dimensions refers to the dimensions of token embedding, MLP Ratio indicates the MLP Ratio in the model, and Depths denotes the number of the transformer layers. In the upper Table, we specify the details of four transformer blocks in Pyramid Vision Transformer for Audio-Visual Segmentation tasks. Each number in the bracket shows the corresponding PVT transformer block's configuration. In the lower Table, we specify the configurations of the transformer layers used for the CAVMAE model. Especially, Depths specifies {the number of modality-specific (unimodal) transformer layers and the number of fusion transformer layers, the total number of transformer layers}.

| Model | # Heads | # Dimensions | MLP Ratio | Depths |
|-------|---------|--------------|-----------|--------|
| PVTv2-b5 | {1, 2, 5, 8} | {64, 128, 320, 512} | {8, 8, 4, 4} | {3, 6, 40, 3} |
| PVTv2-b0 | {1, 2, 5, 8} | {32, 64, 160, 256} | {4, 4, 4, 4} | {2, 2, 2, 2} |

| Model | # Heads | # Dimensions | MLP Ratio | Depths |
|-------|---------|--------------|-----------|--------|
| ViT-Base | 12 | 768 | 4 | {11, 1, 23} |
| ViT-Tiny | 3 | 192 | 4 | {11, 1, 23} |
| ViT-Ex-Tiny | 3 | 192 | 4 | {6, 1, 13} |

- *Self-supervised pre-training* (SSL-FT). The student model is pre-trained with the self-supervised learning (SSL) framework following the training regime for the teacher model (Gong et al., 2022b), and the pre-trained student model is finetuned on the target dataset.

- *Hinton's Knowledge Distillation* (KD) (Hinton et al., 2015). The teacher model's probability distribution from the output logit is distilled to the student model by minimizing the Kullback-Leibler divergence between probability distributions from the teacher and student models.

- *Attention Transfer* (AT) (Zagoruyko & Komodakis, 2016). The student model is trained by distilling spatial attention maps of features at a layer-level. This setting assumes the teacher and the student models have the same number of tokens.

- *Similarity-Preserving Knowledge-Distillation* (SPKD) (Tung & Mori, 2019). The student model is trained by distilling the pairwise similarity between samples from the teacher representation space to the student representation space.

- *Masked Token Similarity Transfer* (MTST) (Choi et al., 2023). The student model is trained by distilling the probability distribution of token similarity. This approach is considered the closest approach to KTD. The main difference between MTST and KTD is teacher information. MTST distills the probability distribution of kernelized tokens, which applies the Softmax function to token similarities, where KTD uses smooth-L1 loss to directly distill the kernelized tokens of the teacher model to the student.

## F  STUDENTS WITH DIFFERENT NUMBERS OF LAYERS

In addition to the results for the ViT-Tiny based student models as presented in the main paper, we include additional results for student models with less number of layers.

We reduce the number of modality-specific layers in the CAVMAE student models from 11 layers to 6 layers, potentially reducing its capacity to learn complex patterns but making it more efficient. This student model configuration reduces the number of parameters from 10M to 8M. We present the results for this student model architecture in Tab. 13. It can be observed that we not only outperform the baseline methods by at least 4% accuracy, we also are able to reach within 3% accuracy of the teacher model while using only 3.4% of the number of teacher model parameters.

## G  FURTHER DETAIL IN ENTROPY CALCULATION ON AVS-BENCH

We clarify the entropy calculation for the teacher's fused feature for the proposed Entropy Monitored Kernelized Token Distillation. Audio-visual segmentation is the task of predicting the mask of the sound source. The fused multimodal feature is directly used to predict the output probability by MLP layer and following sigmoid activation. In this scenario, calculating the average entropy of every

pixel-wise prediction may include unnecessary pixels for the prediction. Here, we clarify that the average entropy is calculated over the ground truth mask region to mitigate the issue.

## H    FUTURE WORKS

While this paper proposes an effective representation to distill a large multi-modal model's knowledge to the small models, we have limited our scope to audio-visual models (Ngiam et al., 2011; Kazakos et al., 2021; Nagrani et al., 2021; Gong et al., 2022b; Huang et al., 2024a; Zhou et al., 2022b; Mao et al., 2023; Gao et al., 2023; Li et al., 2023a; Liu et al., 2024b; 2023a; Yang et al., 2024c; Lin & Bertasius, 2024). We note that our method is broadly applicable beyond audio and visual modalities and can be employed for general compression of multimodal models. Specifically, the main assumption of our work is that the teacher and student models encode the inputs as tokens, e.g., Transformer architectures, which are now widely used. Hence, we see our method being applicable to compress various modalities, both a single modality such as vision (He et al., 2022; Lao et al., 2024a;b; Caron et al., 2021; Duan & Wong, 2026; Fei et al., 2019; Gangopadhyay et al., 2025; Wong & Soatto, 2019; Wang et al., 2026), range (Chen et al., 2025; Huang et al., 2023; Yang et al., 2024b), and tactile (Yang et al., 2024a; 2022), as well as multiple hetereogenous modalities including vision and language (Radford et al., 2021; Li et al., 2024; Wang et al., 2022; 2025; Wu et al., 2025; Zeng et al., 2024a;b), vision and range representations (Chen et al., 2026; Lin et al., 2022; Hu et al., 2021; Park et al., 2020; Yang et al., 2019; Wong et al., 2020; 2021b;a; Wong & Soatto, 2021; Zhu et al., 2021; Wu et al., 2024; Park et al., 2024; 2026; Singh et al., 2023; Park & Jeon, 2024; Chancán et al., 2025; Chung et al., 2025; Zuo et al., 2025; Chen et al., 2024; Ezhov et al., 2024; Rim et al., 2025; 2026; Jeong et al., 2025).

