# OpenReview forum: "Entropy-Monitored Kernelized Token Distillation for Audio-Visual Compression"
_ICLR.cc/2026/Conference — ICLR 2026 Poster_

### Official Review · Reviewer_YgiM · 2025-10-15

**Soundness:** 3
**Presentation:** 2
**Contribution:** 3
**Rating:** 6
**Confidence:** 3

**Summary:**

For network-agnostic audio-visual knowledge distillation, this paper proposes the Entropy-Monitored Kernelized Token Distillation (EM-KTD). Specifically, the Kernelized Token Distillation (KTD) distills pairwise relationships between latent tokens, captured in a Gram matrix. Then, an Entropy-Monitored (EM) scheme is proposed to selectively distill knowledge. It dynamically weighs the distillation loss for each modality (audio, visual, fused) based on the entropy of its feature embeddings. Experiments are conducted on audio-visual event classification and segmentation tasks, demonstrating the effectiveness of the proposed method.

**Strengths:**

- The core idea of KTD is novel and well-motivated. Using kernelization to transfer the geometric structure of the teacher's latent space without requiring feature dimension alignment is an elegant solution to a common problem in knowledge distillation.
- The method is validated on two distinct and challenging audio-visual tasks: classification and segmentation.
- The results are impressive, showing that EM-KTD can compress a teacher model by over 90% while retaining nearly all of its performance, clearly demonstrating the practical utility of the proposed method.

**Weaknesses:**

- The kernelization step has a computational complexity of $O(N^2)$ with respect to the number of tokens $N$ for each instance. While the paper shows strong performance, a more explicit discussion of the training time trade-offs in the main text would be beneficial.
- The paper states that for the Entropy Monitor, "additional task heads... are trained to minimize the cross entropy loss" on the frozen teacher model (Sec. 3.3). This is a crucial implementation detail that lacks clarity. It is unclear *when* these linear probes are trained. Are they pre-trained on the entire dataset before the student distillation process begins? Or are they trained concurrently? A more detailed explanation of this procedure is needed for reproducibility and to fully understand the method's mechanics.
- As shown in Table 2, the EM-KTD is not much superior to KTD in the S4 segmentation task.
- The Figure can be improved. For example, it is challenging for readers to understand the 'entropy-monito' mechanism in Figure 2.

**Questions:**

see weakness

---

> ### Author Response · Authors · 2025-11-21
> **Response to Reviewer YgiM. (Part1)**
>
> Dear Reviewer YgiM,
>
> Thank you for your precious time on the review and your constructive suggestions to improve our manuscript! We appreciate the acknowledgments; for example, the core idea is clear and well-motivated, the results are impressive, and the proposed method has a practical utility.
>
> **W1. The kernelization step has a computational complexity of O($N^2$) with respect to the number of tokens N for each instance. While the paper shows strong performance, a more explicit discussion of the training time trade-offs in the main text would be beneficial.**
>
> We provide discussions on the comparison of the baselines, where all training GPU time are matched across methods in L894 and Fig. 4 in the Supp Mat. The results show that even though the training time is matched, EM-KTD and KTD still shows superior results over existing methods, e.g., KD. Note that training a student model with KTD on VGGSound takes 6.75 hours and KD takes 4.8 hours on a single A100 GPU. We will move this discussion to the main text.
>
> Indeed, to reduce the computational cost for the kernelization, we have implemented a sliding window version of our approach (akin to Swin). This approach kernelizes features within a window shape of $(h/2 \times w/2)$ with the stride of $(h/4,w/4)$ and concatenates them in the batch dimension, reducing the computational cost from $N^2$ to ($\frac{9}{16}N^2$). Using this approach, we found that we are still comparable in performance.
>
>
> **W2a. The paper states that for the Entropy Monitor, "additional task heads... are trained to minimize the cross entropy loss" on the frozen teacher model (Sec. 3.3). This is a crucial implementation detail that lacks clarity. It is unclear when these linear probes are trained.**
>
> The Entropy Monitor for each modality $m$, is a linear layer $g_m(\cdot)$. We will first freeze the teacher model, and the Entropy Monitor will be trained (i.e., linear probing in classification and pixel-wise linear probing in segmentation) before the distillation to the student model. We utilized cosine annealing schedule following the CAVMAE paper to train $g_m(\cdot)$.
>
> We will clarify the training of the Entropy Monitor in the future revision.
>
> **W2b. Are they pre-trained on the entire dataset before the student distillation process begins? Or are they trained concurrently? A more detailed explanation of this procedure is needed for reproducibility and to fully understand the method's mechanics.**
>
> Yes, the teacher model, usually an architecture with a large number of parameters (e.g., ViT-Large, ViT-Giant), is trained on some pretraining dataset before the student model is distilled from it. Then, we train the Entropy Monitor (EM) for each modality on the given training dataset. Finally, the student model is trained from scratch under the guidance of the teacher's output or the teacher's feature representations. The Entropy Monitor assigns a weight to each input instance to lower the influence of the noisy examples during distillation to ensure high-fidelity supervision.
>
> **W3. As shown in Table 2, the EM-KTD is not much superior to KTD in the S4 segmentation task.**
>
> On AVS-S4 dataset, KTD already retains 96\% of the teacher's performance, so the increase (from 96\% to 97\% of EM-KTD) seems small due to saturation in performance.
>
> EM does have a significant effect, as demonstrated by Table 10 of the Supp. Mat., where we applied EM to Hinton's KD. EM improved Hinton's KD from Acc=56.1, mAP=57.3, and mAUC=97.1 to Acc=59.9, mAP=59.7, and mAUC=97.7, which marks a nontrivial improvement (6.71\% on VGG sound). While it may seem like EM has less effect when applied to KTD, this is because KTD already preserves 96.9\% of the teacher model's performance on VGGSound and 97.1\% on S4 benchmarks, which indicates the evaluation metrics are saturated, so it is difficult to push the evaluation metrics further, but we note that EM-KTD still improves the student model consistently over all evaluation metrics.
>
> We note that the computation cost comes from calculating entropy is negligible compared to the cost of computing the kernelization, and it filters out the noisy supervision during the distillation to improve performance downstream.
>
>
> **W4. The Figure can be improved. For example, it is challenging for readers to understand the 'entropy-monito' mechanism in Figure 2.**
>
> We will revise Figure 2 to ensure that Entropy Monitor is clearly illustrated in the next revision.

---

> > ### Comment · Reviewer_YgiM · 2025-11-26
> >
> > Thanks for the authors' rebuttal. Overall, I think the idea is interesting and would like to keep my original rating.

---

> ### Author Response · Authors · 2025-12-03
> **Response to Reviewer YgiM. (Part 2)**
>
> **W1. A follow-up result on sliding-window fashion KTD.**
>
> We provide a detailed number of sliding-window fashion KTD, which consumes only $\times\frac{9}{16}$ of computation compared to the KTD, described in part 1 response in the table below.
>
> |  Method  | Teacher architecure | Student architecure |  Acc | mAP  |  mAUC |
> |:------------:|:--------------:|:------------:|:------:|:------:|:------:|
> | KTD + KD + Sliding | CAVMAE-ViT-Base (164M) | ViT-Tiny (10M) | 61.4 | 62.2 | 97.6 |
> | KTD + KD (Ours) | CAVMAE-ViT-Base (164M) | ViT-Tiny (10M) | 61.4 | 62.3 | 97.6 |
> | EM-KTD + KD + Sliding | CAVMAE-ViT-Base (164M) | ViT-Tiny (10M) | 61.9 | 63.3 | 97.9 |
> | EM-KTD + KD (Ours) | CAVMAE-ViT-Base (164M) | ViT-Tiny (10M) | 62.0 | 63.4 | 97.9 |

---

### Official Review · Reviewer_VPsJ · 2025-10-25

**Soundness:** 3
**Presentation:** 2
**Contribution:** 3
**Rating:** 8
**Confidence:** 3

**Summary:**

This paper presents Entropy-Monitored Kernelized Token Distillation (EM-KTD), a novel framework for compressing audio-visual models. The method features two key components: 1) Kernelized Token Distillation (KTD), which distills the pairwise relationships between latent tokens captured in a Gram matrix, making the approach architecture-agnostic and highly expressive. 2) An Entropy-Monitored (EM) scheme that adaptively weights each modality's contribution based on its predictive entropy, ensuring high-fidelity supervision. This framework establishes a new state-of-the-art on two audio-visual tasks.

**Strengths:**

1.The paper proposes a novel EM-KTD framework. By operating on the space of token relationships, it enables flexible and architecture-agnostic knowledge transfer.

2.The method demonstrates state-of-the-art performance on audio-visual benchmarks, achieving a 94% parameter reduction while retaining over 96% of the teacher's performance.

3.The claims are well-supported by comprehensive comparative experiments and thorough ablation studies that validate the contributions of each component.

**Weaknesses:**

1.The O(N²) computational complexity of the pairwise kernelization step may limit the method's scalability to tasks with more input tokens.

2.The EM scheme relies on entropy-loss-based classification tasks. This might limit its direct applicability to other tasks (e.g. regression tasks).

3.The paper's validation on heterogeneous teacher-student architectures is a key strength, but this experimental context is detailed only in the appendix. Highlighting this setup in the main text would better frame the results and underscore the method's flexibility. To provide a more comprehensive evaluation, an additional experiment in a homogeneous setting is recommended to evaluate the method's performance when architectures are matched.

Typos:

1. The formulation of the Huber loss in Equation (2) has two minor errors: it is missing an equals sign, and the condition should be ||p - q|| < 1 instead of ||p, q|| < 1.

**Questions:**

See weaknesses.

---

> ### Author Response · Authors · 2025-11-21
> **Response to Reviewer VPsj. (Part1)**
>
> Dear reviewer VPsj,
>
> We appreciate your precious time and effort on the review and your constructive suggestions to improve our manuscript! We appreciate the acknowledgments; for example, this paper proposes a novel framework (EM-KTD), the model demonstrates the state-of-the-art result (94\% parameter reduction, preserving 96\% of the performance) ,the claims are well-supported by comprehensive comparative experiments and thorough ablation studies that validate the contributions of each component.
>
>
> **W1. The O($N^2$) computational complexity of the pairwise kernelization step may limit the method's scalability to tasks with more input tokens.**
>
> Indeed, like operations such as attention, computing a kernel is O($N^2$). We note that masking applied to MTST does not resolve this issue, as their implementation does not remove tokens before computing the cosine similarity between the features. Also, to reduce the computational cost for the kernelization, we have tried a sliding window approach. This approach kernelizes features within a window shape of $(h/2 \times w/2)$ with the stride of $(h/4,w/4)$ (akin to Swin Transformers) and concatenate them in the batch dimension, reducing the computational cost from $N^2$ to ($\frac{9}{16}N^2$), and found that we are still comparable in performance.
>
>
> **W2. The EM scheme relies on entropy-loss-based classification tasks. This might limit its direct applicability to other tasks (e.g. regression tasks).**
>
> While we focus on classification-based tasks in this paper, EM is not limited to them. In the case of regression tasks, we can re-formulate the objective to estimate uncertainty instead. Specifically, we can consider variance-based uncertainty such as Gaussian negative log-likelihood loss:
>
> \begin{equation}
> \mathcal{L} = \frac{1}{2}\left(\frac{(y - \mu)^2}{\sigma^2} + \log \sigma^2 \right).
> \end{equation}
>
> where we will employ a regression head and an uncertainty head, which will jointly minimize this objective.
>
> **W4. Typos:
> The formulation of the Huber loss in Equation (2) has two minor errors: it is missing an equals sign, and the condition should be ||p - q|| < 1 instead of ||p, q|| < 1.**
>
> Thank you, we will fix them accordingly.

---

> ### Author Response · Authors · 2025-12-03
> **Response to Reviewer VPsj. (Part 2)**
>
> **W1. A follow-up result on sliding-window fashion KTD.**
>
> We provide a detailed number of sliding-window fashion KTD, which consumes only $\times\frac{9}{16}$ of computation compared to the KTD, described in part 1 response in the table below.
>
> |  Method  | Teacher architecure | Student architecure |  Acc | mAP  |  mAUC |
> |:------------:|:--------------:|:------------:|:------:|:------:|:------:|
> | KTD + KD + Sliding | CAVMAE-ViT-Base (164M) | ViT-Tiny (10M) | 61.4 | 62.2 | 97.6 |
> | KTD + KD (Ours) | CAVMAE-ViT-Base (164M) | ViT-Tiny (10M) | 61.4 | 62.3 | 97.6 |
> | EM-KTD + KD + Sliding | CAVMAE-ViT-Base (164M) | ViT-Tiny (10M) | 61.9 | 63.3 | 97.9 |
> | EM-KTD + KD (Ours) | CAVMAE-ViT-Base (164M) | ViT-Tiny (10M) | 62.0 | 63.4 | 97.9 |
>
> **W3a. The paper's validation on heterogeneous teacher-student architectures is a key strength, but this experimental context is detailed only in the appendix. Highlighting this setup in the main text would better frame the results and underscore the method's flexibility.**
>
> We will revise the manuscript to highlight this strength in the main text.
>
>
> **W3b. To provide a more comprehensive evaluation, an additional experiment in a homogeneous setting is recommended to evaluate the method's performance when architectures are matched.**
>
> We provide the results with a homogeneous architecture where the teacher and student architectures are matched in the table below.
>
> |  Method  | Teacher architecure | Student architecure |  Acc | mAP  |  mAUC |
> |:------------:|:--------------:|:------------:|:------:|:------:|:------:|
> | EM-KTD + KD (Ours) | CAVMAE-ViT-Base (164M) | ViT-Tiny (10M) | 62.0 | 63.4 | 97.9 |
> | EM-KTD + KD (Ours) | CAVMAE-ViT-Base (164M) | ViT-Base (164M) | 64.1 | 65.4 | 98.0 |

---

### Official Review · Reviewer_rDf2 · 2025-10-30

**Soundness:** 4
**Presentation:** 3
**Contribution:** 2
**Rating:** 6
**Confidence:** 3

**Summary:**

The paper proposed EM-KTD, a novel framework to compress audio-visual models with knowledge distillation. The proposed method expands the idea of MTST (i.e., distilling pairwise relationships between token embeddings) to audio-visual understanding models, and makes the following improvements:

1. Measures the similarity between latent embeddings with kernel functions rather than cosine similarity.
2. Replaces KL-divergence loss with Huber loss, eliminating the need to use masking and Softmax mapping as in [MTST.](http://MTST.In) In this way, the student model can replicate the geometry of teacher latent space more precisely.
3. Proposes Entropy Monitor to adjust the loss weight of each modality entropy, based on entropy of uni-modal classification predictions.

The authors evaluated the proposed method on two audio-visual understanding tasks: event classification and segmentation. Experiment shows a clear advantage of EM-KTD over both vanilla training and previous distillation baselines, providing an resource-efficient solution for audio-visual understanding.

**Strengths:**

1. The paper proposed an effective solution to the latent dimension mismatch problem in audio-visual knowledge distillation.
2. Reasons for core design choices are clearly explained evidently supported by experiments, such as why removing softmax mapping (Appendix A) and why distilling token relationships within one instance in each modality (Appendix B).
3. The proposed model is architecture-agnostic and supports various kernel functions (linear, polynomial, RBF), allowing trade-offs between computation complexity and performance.

**Weaknesses:**

1. Compatibility to long training samples: Calculating the Gram matrix is an $O(N^2)$ operation, which may be computationally inefficient, especially when masking is not applied as in MTST. This might affect effectiveness of the proposed method in modalities with variable input lengths, like audio.
2. While the advantage of KTD over MTST is concrete, the advantage of EM-KTD over KTD is comparably less significant.
3. Figure 2 is a bit confusing. The “fusion modality” are not depicted and mechanism of entropy-weighted loss is ambiguous.

**Questions:**

Lacking Multi-modal Distillation Baselines: Although EM-KTD is designed for audio-visual knowledge distillation, all baselines included are not specifically designed for multi-modal knowledge distillation. Can the proposed method be compared with multi-modal knowledge distillation methods mentioned in Related Work?

---

> ### Author Response · Authors · 2025-11-21
> **Response to Reviewer rDf2. (Part1)**
>
> Dear Reviewer rDf2,
>
> Thank you for your precious time on the review and your constructive suggestions to improve our manuscript! We appreciate the acknowledgments; for example, the paper proposed an effective solution. The reasons for core design choices are clearly explained, and the proposed model is architecture-agnostic and supports various kernel functions.
>
> Responses below are marked with Q\# for Question and W\# for Weakness.
>
> **W1. Compatibility to long training samples: Calculating the Gram matrix is an O($N^2$) operation, which may be computationally inefficient, especially when masking is not applied as in MTST. This might affect effectiveness of the proposed method in modalities with variable input lengths, like audio.**
>
> Indeed, like operations such as attention, computing a kernel is O($N^2$). We note that masking applied to MTST does not resolve this issue, as their implementation does not remove tokens before computing the cosine similarity between the features. Also, to reduce the computational cost for the kernelization, we have tried a sliding window approach. This approach kernelizes features within a window shape of $(h/2 \times w/2)$ with the stride of $(h/4,w/4)$ (akin to Swin Transformers) and concatenate them in the batch dimension, reducing the computational cost from $N^2$ to ($\frac{9}{16}N^2$), and found that we are still comparable in performance.
>
> **W2. While the advantage of KTD over MTST is concrete, the advantage of EM-KTD over KTD is comparably less significant.**
>
> We respectfully disagree. EM does have a significant effect as demonstrated by Table 10 of Supp. Mat., where we applied EM to Hinton's KD. EM improved Hinton's KD from Acc=56.1, mAP=57.3, and mAUC=97.1 to Acc=59.9, mAP=59.7, and mAUC=97.7, which marks a nontrivial improvement (6.71\% on VGG sound). While it may seem like EM has less effect when applied to KTD, this is because KTD already preserves 97.6\% of the teacher model's performance, which indicates the evaluation metrics saturated, so it is difficult to push the evaluation metrics further, but we note that EM-KTD still improves the student model consistently over all evaluation metrics.
>
> Also. the additional computation cost of EM comes from calculating entropy is negligible compared to the cost of computing the kernelization, and it filters out the noisy modality during the distillation.
>
> **W3. Figure 2 is a bit confusing. The “fusion modality” are not depicted and mechanism of entropy-weighted loss is ambiguous.**
>
> The fusion modality was omitted from the figure for the simplicity, and we will add it back to Fig. 2 in the next revision.

---

> ### Author Response · Authors · 2025-12-03
> **Response to Reviewer rDf2. (Part 2)**
>
> **W1. a follow-up result**
>
> We provide a detailed number of sliding-window fashion KTD here.
>
> |  Method  | Teacher architecure | Student architecure |  Acc | mAP  |  mAUC |
> |:------------:|:--------------:|:------------:|:------:|:------:|:------:|
> | KTD + KD + Sliding | CAVMAE-ViT-Base (164M) | ViT-Tiny (10M) | 61.4 | 62.2 | 97.6 |
> | KTD + KD (Ours) | CAVMAE-ViT-Base (164M) | ViT-Tiny (10M) | 61.4 | 62.3 | 97.6 |
> | EM-KTD + KD + Sliding | CAVMAE-ViT-Base (164M) | ViT-Tiny (10M) | 61.9 | 63.3 | 97.9 |
> | EM-KTD + KD (Ours) | CAVMAE-ViT-Base (164M) | ViT-Tiny (10M) | 62.0 | 63.4 | 97.9 |
>
>
> **Q1. Lacking Multi-modal Distillation Baselines: Although EM-KTD is designed for audio-visual knowledge distillation, all baselines included are not specifically designed for multi-modal knowledge distillation. Can the proposed method be compared with multi-modal knowledge distillation methods mentioned in Related Work?**
>
> Yes. The result with the audio-visual compression method in the related work section (MOMA) on VGGSound is provided in the table below.
>
> |  Method  | Teacher architecure  | Student architecure  |  Acc | mAP  |  mAUC |
> |:------------:|:--------------:|:------------:|:------:|:------:|:------:|
> | MOMA | CAVMAE-ViT-Base (164M) | ViT-Tiny (10M) | 58.9 | 60.3 | 97.2 |
> | EM-KTD + KD (Ours) | CAVMAE-ViT-Base (164M) | ViT-Tiny (10M) | 62.0 | 63.4 | 97.9 |

---

### Official Review · Reviewer_6Yud · 2025-10-30

**Soundness:** 3
**Presentation:** 2
**Contribution:** 2
**Rating:** 4
**Confidence:** 3

**Summary:**

Traditional knowledge distillation methods either require alignment of the teacher-student structure or fail to effectively utilize the internal structural information of modalities, and they lack an adaptive mechanism for the varying amounts of information across different modalities. To address these issues, the paper proposes Kernelized Token Distillation (KTD): instead of distilling the features themselves, it distills the similarity structure among tokens within a single sample, achieving structure-agnostic cross-modal distillation. Additionally, the paper introduces the Entropy-Monitored mechanism: by using the classification entropy of each modality from the teacher model to dynamically adjust the distillation weights and suppress the interference of low-information modalities. The paper validates the effectiveness on the VGGSound and AVS-Bench datasets, maintaining a considerable level of teacher performance even when the student model has significantly fewer parameters.

**Strengths:**

The EM-KTD method uses the "structure among tokens" as the distillation target, avoiding the dimension matching problem. The motivation for the entropy supervision mechanism is reasonable, and the overall method is innovative. The clarity of the paper is acceptable, but it lacks intuitive explanations for "kernelized token" and "Gram matrix." It is suggested to optimize Figure 1 and Figure 2 to emphasize that the similarity matrix comes from multiple tokens within a single sample. The experimental quality of the paper is solid, with comprehensive comparisons to current mainstream methods and extensive ablation experiments. As a general method, if the paper could validate it on more modalities and tasks, it would further enhance its significance.

**Weaknesses:**

1. The paper's explanation of the entropy prediction head $g_m(⋅)$ is insufficient. Firstly, in the methods section, it does not describe which stage of the process the entropy predictor is trained in. In the experimental section, it does not mention the source of the weights for $g_m(⋅)$. In the ablation part, it seems that the impact of the structure of $g_m(⋅)$ on the distillation results is not discussed.

2. The explanation of dataset labels and information on line 192 is not clear enough. Clarifying the meanings of $n$ and $N$ in the Dataset section would make the paper more understandable.

3. Validating the method on more datasets would more fully demonstrate its generalizability.

**Questions:**

1. I still have some confusion about the entropy monitor $g_m(⋅)$. Firstly, how is it trained? If the task is not classification, how should the model evaluate the entropy?

2. Although the paper tries linear, polynomial, and RBF kernel functions, it does not provide a systematic selection criterion or adaptive mechanism. When used for different tasks or datasets, is it necessary to manually select the kernel function?

3. The paper mainly focuses on the visual and audio modalities. I am curious whether, as a general method, it would yield similar results on tasks involving other modalities (such as visual-text, audio-text, etc.). If this could be validated, it would well demonstrate the method's versatility.

---

> ### Author Response · Authors · 2025-11-21
> **Response to Reviewer 6Yud. (Part1)**
>
> Dear Reviewer 6Yud,
>
> Thank you for your precious time on the review and your constructive suggestions to improve our manuscript! We appreciate the acknowledgments; for example, the experimental quality of the paper is solid. The motivation for the entropy supervision mechanism is reasonable, and the overall method is innovative.
>
> Responses below are marked with Q# for Question and W# for Weakness.
>
> **W1. The paper's explanation of the entropy prediction head $g_m(\cdot)$ is insufficient. Firstly, in the methods section, it does not describe which stage of the process the entropy predictor is trained in. In the experimental section, it does not mention the source of the weights for $g_m(\cdot)$. In the ablation part, it seems that the impact of the structure of $g_m(\cdot)$ on the distillation results is not discussed.**
>
> The linear layer for each modality $m$, $g_m(\cdot)$, has been trained before the distillation, after the teacher model has been frozen (i.e., linear probing in classification and pixel-wise linear probing in segmentation). We utilized cosine annealing schedule following the CAVMAE paper to train $g_m(\cdot)$.
>
> **W2. The explanation of dataset labels and information on line 192 is not clear enough. Clarifying the meanings of n and N in the Dataset section would make the paper more understandable.**
>
> $N$ refers to the size of the dataset, $n$ denotes the index of an example within the dataset.
>
> **Q1. I still have some confusion about the entropy monitor $g(\cdot)$. Firstly, how is it trained? If the task is not classification, how should the model evaluate the entropy?**
>
> We considered single- and multisound source segmentation (on AVS-Bench), and audio-visual event classification
> (on VGGSound). In both classification and segmentation tasks, we have access to classification labels and segmentation masks; therefore, we use them to train the entropy monitors $g(\cdot)$ (classification and segmentation heads) for image, audio, and fused features by minimizing cross entropy. In the case of regression tasks, where the probability distribution cannot be utilized, we can re-formulate the objective to estimate uncertainty instead. Specifically, we can consider variance-based uncertainty such as Gaussian negative log-likelihood loss:
>
> \begin{equation}
> \mathcal{L} = \frac{1}{2}\left(\frac{(y - \mu)^2}{\sigma^2} + \log \sigma^2 \right).
> \end{equation}
>
> where we will employ a regression head and an uncertainty head, which will jointly minimize this objective.
>
> **Q2: Although the paper tries linear, polynomial, and RBF kernel functions, it does not provide a systematic selection criterion or adaptive mechanism. When used for different tasks or datasets, is it necessary to manually select the kernel function?**
>
> We do provide a selection criterion based on fidelity and reported that the RBF kernel is preferred based on this criterion, as seen in Table 3 of the main paper. We do note that there is some overhead in choosing RBF kernel (see Eq. 5), which we discussed as trade-offs between computation and performance in Sec. 4.2.
>
> **Q3. The paper mainly focuses on the visual and audio modalities. I am curious whether, as a general method, it would yield similar results on tasks involving other modalities (such as visual-text, audio-text, etc.). If this could be validated, it would well demonstrate the method's versatility.**
>
> Yes, while our focus is on visual and audio, we do have preliminary experiments on visual-text, which demonstrates the generalizability of our method; we do note that this is beyond the scope of our work.

---

> ### Author Response · Authors · 2025-12-03
> **Response to Reviewer 6Yud. (Part 2)**
>
> **W1. Follow-up on the additional results with the different structure of $g_m(\cdot)$.**
>
> We observe a marginal impact on the performance. The MLP structure consists of the linear layers with non-linear activations (ReLU), with the hidden size of input dimension$\times 2$.
>
> This indicates a single linear layer as $g_m(\cdot)$ is sufficient to monitor an entropy.
>
> |  Method  | Teacher architecure | Student architecure |  Acc | mAP  |  mAUC |
> |:------------:|:--------------:|:------------:|:------:|:------:|:------:|
> | EM-KTD + KD + 3layer MLP | CAVMAE-ViT-Base (164M) | ViT-Tiny (10M) | 61.7 | 62.7 | 97.6 |
> | EM-KTD + KD + 2layer MLP | CAVMAE-ViT-Base (164M) | ViT-Tiny (10M) | 62.0 | 63.3 | 97.9 |
> | EM-KTD + KD + 1layer Linear (Ours) | CAVMAE-ViT-Base (164M) | ViT-Tiny (10M) | 62.0 | 63.4 | 97.9 |
>
>
> **W3. Validating the method on more datasets would more fully demonstrate its generalizability.**
> and
> **Q3. The paper mainly focuses on the visual and audio modalities. I am curious whether, as a general method, it would yield similar results on tasks involving other modalities (such as visual-text, audio-text, etc.). If this could be validated, it would well demonstrate the method's versatility.**
>
> Yes, while our focus is on visual and audio, we do have preliminary experiments on the visual-text (open-vocabulary segmentation), which demonstrate the generalizability of our method; we do note that this is beyond the scope of our work. The table below shows additional results on the open-vocabulary segmentation (LSeg) on ADE 20k. We improved Hinton's KD by 4.1\% on pixel-wise accuracy and 11.9\% on mIOU.
>
>
> |       Model     | Training   |  Vision encoder  |  Text encoder  | pixel-acc | mIOU |
> |:------------:|:------:|:------:|:------:|:------:|:------:|
> | Lseg-teacher  | Supervised | ViT-L/16 | ViT-B/32 | 79.2 | 28.1 |
> | Lseg-student  | Supervised | ViT-B/32 | ViT-B/32 | 71.7 | 21.1 |
> | Lseg-student  | Hinton's KD | ViT-B/32 | ViT-B/32| 73.1 | 23.5 |
> | Lseg-student  | EM-KTD | ViT-B/32 | ViT-B/32 | 76.1 | 26.3 |

---

### Author Response · Authors · 2025-12-03
**Summary of the Rebuttal Period for Area Chair**

Dear Handling Area Chair,

We sincerely thank all reviewers and the Area Chair for their thorough reviews and constructive feedback. We are grateful for the time and effort invested in evaluating and discussing our work. We thank the reviewers' acknowledgment that: (1) The core design choices are well explained (rDf2) and solve the dimension mismatch problem (6Yud, rDf2, YgiM). (2) The method is well-motivated (YgiM, rDf2, 6Yud). (3) The result is impressive (YgiM), the experimental quality is solid (6Yud, VPsj). (4) The ablations are well-supported (rDf2 and VPsj). (5) The strength in our proposed method being architecture-agnostic (rDf2) and (6) supports various kernel functions (rDf2).

Our response addressed each point raised by the reviewers. While we did not get a chance to interact with rDf2, VPsj, and 6Yud, we note that YgiM did respond and kept the rating for acceptance after the rebuttal. The update from YgiM was completed by 14:15 UTC, 26 November 2025, which predates the public disclosure of the recent information leak (widely reported between 14:00 and 16:00 UTC on 27 November 2025).

For your reference, we summarize our rebuttal efforts, including our responses to the reviewers' concerns, the additional clarifications and revisions we provided, and the subsequent results added during the discussion phase. Our key revisions include:

(1) Additional results to answer the reviewers.

**Results with various $g_m(\cdot)$ architectures (6Yud).**

We show results (6Yud, W1 in Part 2) with the various MLP architectures for the entropy monitor ($g_m(\cdot)$), which demonstrates the current design choice of $g_m$ with a single linear to be sufficient.

**Additional result on the visual-text task. (6Yud).**

We provided the result on a visual-text task, open-vocabulary segmentation, where our EM-KTD shows the state-of-the-art performance compared to Hinton's KD in the response (6Yud, W3 in Part 2).

**The sliding-window fashion KTD to reduce the computation of kernelization (rDf2, VPsj, YgiM).**

We provided an additional result of KTD with a lighter version of the kernelization, following the Swin transformer. The result shows that the sliding-window kernelization not only can save the computation but also preserves the performance. The results are shown in the responses to rDf2, VPsj, and YgiM, W1 in Part 1.

**An additional audio-visual method from related work on VGGSound (rDf2).**

We compared an additional knowledge distillation method on the audio-visual task, MOMA, to extend the baselines (rDf2, Q1 in Part 2).

(2) Important questions from the reviewers.

**The details on the entropy monitor $g_m(\cdot)$ (6Yud, YgiM)**

We provided explanations on $g_m(\cdot)$, regarding the training and how they are used during the knowledge distillation process. (6Yud, W1, Q1 in Part 1, YgiM, W2a and W2b in Part 1).

**The EM scheme relies on entropy-loss-based classification tasks. How to do EM in the other tasks (e.g., regression tasks)? (6Yud and VPsj)**

We provide a further explanation on predicting log Gaussian uncertainty in place of entropy for regression tasks (6Yud, Q1 in Part 1, and VPsj, W2 in Part 1).

**EM-KTD is not much superior to KTD (YgiM, rDF2)**

We provided an explanation in the response to YgiM (W3 in Part 1) and rDF2 (W2 in Part 1): KTD already retains 96\% of the teacher's performance, so the increase (from 96\% to 97\%) seems small due to saturation in performance. The contributions of EM is significant as demonstrated in Tab. 10 of Supp. Mat., where applying EM to Hinton's KD resulted in a larger boost. We note that EM-KTD consistently improves KTD across all experiments.

(3) Contents in the paper.

We will revise the contents itemized below.

- The typos in Eq. 2 (VPsj, W4 in Part 1).

- Fig. 2 should include the fusion modality (rDF2, W3 in Part 1).

- Typo in Fig. 2 (entropy-monitor) (YgiM, W4 in Part 1).

All contents discussed during the rebuttal will be incorporated into the next revision of the manuscript.

Thank you again for your precious time and efforts during review and rebuttal.

Paper 4654 Authors

---

### Meta-Review · Area_Chair_As9F · 2025-12-22

**Summary:**

Reviewer 6Yud argued that the clarification of entropy prediction head and dataset labels are not clearly described. Reviewer rDf2, Reviewer VPsJ and Reviewer YgiM expressed their concerns on inefficient training for long samples due to Gram matrix.

AC find that the idea is very similar to relational knowledge distillation (RKD) [R1], which distills knowledge by the pairwise distance loss. The authors should clearly compare their method with RKD on both related work and experiments.

[R1] Park et al. Relational Knowledge Distillation. CVPR 2019.

**Reviewer Concerns:**

The explanation about entropy prediction head and dataset labels are well addressed. The issue about inefficient training for long samples doesn't affect the contributions of this paper. AC suggests that authors should well compare their method with RKD.

[R1] Park et al. Relational Knowledge Distillation. CVPR 2019.

**Reviewer Scores:**

Reviewer 6Yud: 4;

Reviewer rDf2: 6;

Reviewer VPsJ: 8;

Reviewer YgiM: 6

---

### Decision · Program_Chairs · 2026-01-26

Accept (Poster)